# A genome-centric view of the role of the *Acropora kenti* microbiome in coral health and resilience

**Lauren F. Messer** [1,2] ✉, **David G. Bourne** [3,4], **Steven J. Robbins**[5], **Megan Clay** [1], **Sara C. Bell** [3,4], **Simon J. McIlroy** [1] & **Gene W. Tyson** [1] ✉

Microbial diversity has been extensively explored in reef-building corals. However, the functional roles of coral-associated microorganisms remain poorly elucidated. Here, we recover 191 bacterial and 10 archaeal metagenome-assembled genomes (MAGs) from the coral *Acropora kenti* (formerly *A. tenuis*) and adjacent seawater, to identify microbial functions and metabolic interactions within the holobiont. We show that 82 MAGs were specific to the *A. kenti* holobiont, including members of the Pseudomonadota, Bacteroidota, and Desulfobacterota. *A. kenti*-specific MAGs displayed significant differences in their genomic features and functional potential relative to seawater-specific MAGs, with a higher prevalence of genes involved in host immune system evasion, nitrogen and carbon fixation, and synthesis of five essential B-vitamins. We find a diversity of *A. kenti*-specific MAGs encode the biosynthesis of essential amino acids, such as tryptophan, histidine, and lysine, which cannot be de novo synthesised by the host or Symbiodiniaceae. Across a water quality gradient spanning 2° of latitude, *A. kenti* microbial community composition is correlated to increased temperature and dissolved inorganic nitrogen, with corresponding enrichment in molecular chaperones, nitrate reductases, and a heat-shock protein. We reveal mechanisms of *A. kenti*-microbiome-symbiosis on the Great Barrier Reef, highlighting the interactions underpinning the health of this keystone holobiont.

Microorganisms underpin the productivity of coral reefs[1,2] and act as sensitive indicators of ecosystem health[3]. Indeed, the ecological success of corals throughout nutrient-poor tropical marine waters globally has been attributed to the symbiotic relationship between reef-building corals and unicellular algae of the family Symbiodiniaceae[4]. Extensive phylogenetic surveys of corals over the last ~20 years have identified hundreds to thousands of other distinct microorganisms, including bacteria, archaea, fungi, protists, and their viruses, which collectively represent the coral holobiont[5,6]. Within the coral holobiont framework, the microbiome is hypothesised to play key roles in nutrient cycling[7–10], antioxidant production[11], and in defence against pathogens and disease progression[12]. However, previous studies have largely relied upon culture-dependent characterisation[13], gene-centric metagenomics[14], or metabolic predictions based on taxonomy derived from 16S rRNA gene sequencing[15], which may not accurately capture the lineage-specific and community-level functions of coral-associated

[1]Centre for Microbiome Research, School of Biomedical Sciences, Translational Research Institute, Queensland University of Technology, Brisbane, QLD 4102, Australia. [2]Division of Biological and Environmental Sciences, Faculty of Natural Sciences, University of Stirling, Stirling FK9 4LA Scotland, UK. [3]College of Science and Engineering, James Cook University, Townsville, QLD 4811, Australia. [4]Australian Institute of Marine Science, Townsville, QLD 4810, Australia. [5]Australian Centre for Ecogenomics, School of Chemistry and Molecular Biosciences, The University of Queensland, Brisbane, QLD 4072, Australia. ✉e-mail: lauren.messer@stir.ac.uk; gene.tyson@qut.edu.au

microorganisms. The use of genome-resolved metagenomics to elucidate microbial functional roles remains challenging due to relatively low microbial biomass and the close associations between holobiont cells[16], resulting in overwhelming host DNA signals[17]. Consequently, although microbial diversity has been widely explored in reef-building corals, the specific functions of holobiont microorganisms and the metabolic interactions between them are poorly resolved relative to other host-associated systems.

To overcome this limitation, methods designed to enrich tissue-associated microbial cells for metagenomic sequencing recently enabled the comprehensive analysis of coral holobiont functional potential for the massive coral *Porites lutea*[18]. Using this approach, 52 bacterial and archaeal metagenome-assembled genomes (MAGs) and a *Cladocopium* C15 sp. genome were resolved. Metabolic reconstruction revealed the dependence of *P. lutea* and *Cladocopium* C15 sp. on the microbiome for essential B-vitamin production and provided genomic evidence for holobiont recycling of methylated sulfur molecules[18]. In addition, further analysis revealed that the *P. lutea* microbiome can produce osmolytes such as glycine betaine, and that this may occur across a range of coral host species[19]. Whether these findings represent the underlying features of coral-microbiome symbiosis remains to be addressed and will require further investigations of the functional potential of natural microbial populations within the coral holobiont.

The common branching coral *Acropora kenti* (formerly *A. tenuis*[20]), which is widely distributed across the Great Barrier Reef (GBR), represents an ideal coral species to further refine models for coral-host-microbiome symbioses. Historically, *A. kenti* was considered an environmentally sensitive coral species, negatively impacted by nutrient enrichment and thermal stress[21], and has consequently been the subject of active reef restoration initiatives which demonstrate the importance of symbiotic Symbiodiniaceae in *A. kenti* holobiont resilience (e.g.,[22,23]). Importantly, the genome of this coral, and that of the dominant Symbiodiniaceae *Cladocopium goreaui*, have recently been sequenced[24–26], which facilitates the in silico separation of host and microbial DNA sequences for genome-resolved metagenomic analysis of the microbiome. Previously, 16S rRNA gene sequencing has revealed that *A. kenti* harbours colony-specific microbial communities that display signatures of adaptation to their local environment[27]. Stable isotope analysis indicates that members of the *Acropora* genus are largely autotrophic, such that *A. kenti* primarily depends upon Symbiodiniaceae for essential nutrients, as opposed to acquisition through heterotrophic feeding[28]. Taken together, we hypothesise that the *A. kenti* microbiome likely plays a key role in nutrient provisioning, in addition to Symbiodiniaceae, and may support the holobionts' ecological success across the GBR. Collectively, recent advancements now enable the detailed characterisation of *A. kenti* microbial functional roles to address this hypothesis and place important biological functions encoded by the microbiome within their genomic and phylogenetic context.

Recently, genome-wide analyses of *A. kenti* and *Cladocopium* have revealed their genetic diversity across the GBR[24,29]. *A. kenti* colonies appear as distinct populations at coastal sites along the inshore GBR yet maintain consistency in their dominant *Cladocopium* sp. across different locations[24]. Geographic patterns in *A. kenti* sympatric genetic clusters and *Cladocopium* genetic diversity appear to be underpinned by environmental water quality gradients[24,29], yet whether consistent patterns are displayed in the *A. kenti* microbiome and their functions remains to be addressed. Here, visually healthy colonies from six locations on the GBR, which spanned 2° of latitude from the northernmost to southernmost site, were sampled for genome-resolved metagenomics to elucidate microbial functional roles. The variability in environmental water quality between these sites, which included two (one northern and one southern) gradients of river-plume impacted (i.e., poor water quality) to less impacted reefs[24,30], provided a natural laboratory to explore the factors influencing *A. kenti*

microbial community structure and metabolic potential. Our findings demonstrate that key molecular mechanisms encoded by the microbiome may underpin the healthy functioning of the *A. kenti* holobiont, supporting their present-day ecological success and likely influencing their fate in future oceans.

## Results and discussion

### Recovery of MAGs from *A. kenti*

A combination of microbiome-enrichment, for the concentration of microorganisms associated with coral tissues and mucus, and metagenomics (118 Gbp post-QC, see Methods; Supplementary Fig. 1) was used to recover an unprecedented number of coral-associated MAGs from *A. kenti* of the inshore GBR ($n = 22$). In total, 170 *A. kenti* MAGs displaying quality scores ≥50 (completeness – 3 × contamination[18]) were identified, resulting in 102 'medium' to 'high' quality MAGs ($91.2 \pm 7.12\%$ mean completeness, and $2.03 \pm 2.16\%$ mean contamination), following dereplication with additional filtering. These MAGs spanned 12 bacterial and 1 archaeal phyla including genomes from members of the Pseudomonadota ($n = 46$), Cyanobacteriota ($n = 18$), Bacteroidota ($n = 14$), Desulfobacterota ($n = 7$), Verrucomicrobiota ($n = 5$), Actinobacteriota ($n = 3$), Bacillota ($n = 3$), Chlamydiota ($n = 1$), Desulfobacterota_D ($n = 1$), Bacillota_A ($n = 1$), Nitrospirota ($n = 1$), Planctomycetota ($n = 1$), and Thermoproteota ($n = 1$), representing 60 families and 87 genera (Fig. 1a, b; Supplementary Data 1). This is consistent with previous 16S rRNA gene sequencing studies which have identified these lineages in *A. kenti*[31] and are also in-line with the dominance of these phyla represented by cultured isolates of coral-associated bacteria more broadly[13]. However, MAGs from novel putative intracellular coral symbionts including Chlamydiota (*Simkania* sp.), Bacillota (*Mycoplasmataceae*), and non-photosynthetic Cyanobacteriota of the order Gastranaerophilales, which are rarely observed within scleractinian corals[32–34], were also identified. In samples collected from the surrounding seawater ($n = 6$), which represented a free-living microbial community 'control' for phylogenetic and functional comparisons, 182 MAGs were recovered, with 99 MAGs remaining following dereplication ($86.7 \pm 7.25\%$ mean completeness, and $2.89 \pm 2.31\%$ mean contamination; Fig. 1a, b; see Supplementary Note 1). The identified MAGs were estimated to represent 54% and 32% of microbial genera across the *A. kenti* and seawater communities, respectively (Fig. 1c). Investigation of the prevalence and relative abundances of these MAGs demonstrated their heterogeneity across the two sample types (Fig. 2a) and indicated that 20 of the *A. kenti*-derived MAGs were more abundant within the surrounding seawater (hereafter, seawater-specific; Fig. 2b). While the remaining 82 *A. kenti* MAGs were specific to the *A. kenti* holobiont (hereafter, *A. kenti*-specific; Fig. 2c; see Supplementary Note 2). Among these, Pseudomonadota dominated in all but one of the *A. kenti* samples (Russell Island replicate 2), and collectively comprised up to ~90% of the total mapped reads (relative abundance) of MAGs in each sample. The relative abundances of individual *A. kenti*-specific MAGs ranged from a minimum of ~0.01% (e.g., novel Flavobacteriaceae MAG, Magnetic_MAG16), to a maximum of 70.5% of mapped reads (e.g., *Chlorobium_A marina*, Russell_MAG40), with all MAGs displaying site specific differences in relative abundance. Collectively, the relatively high recovery of genus-level representatives (Fig. 1c) specific to *A. kenti* and consistency of the MAGs with previous literature, indicated that many of the key microbial members of the *A. kenti* colonies were recovered following microbiome enrichment, allowing metabolic reconstruction of the *A. kenti* microbiome.

### *A. kenti*- and seawater-specific MAGs display distinct genomic features

Statistical comparison of the *A. kenti*- and seawater-specific MAGs genomic features revealed distinct characteristics and functions providing insight into the coral-associated ecological niche. Overall,

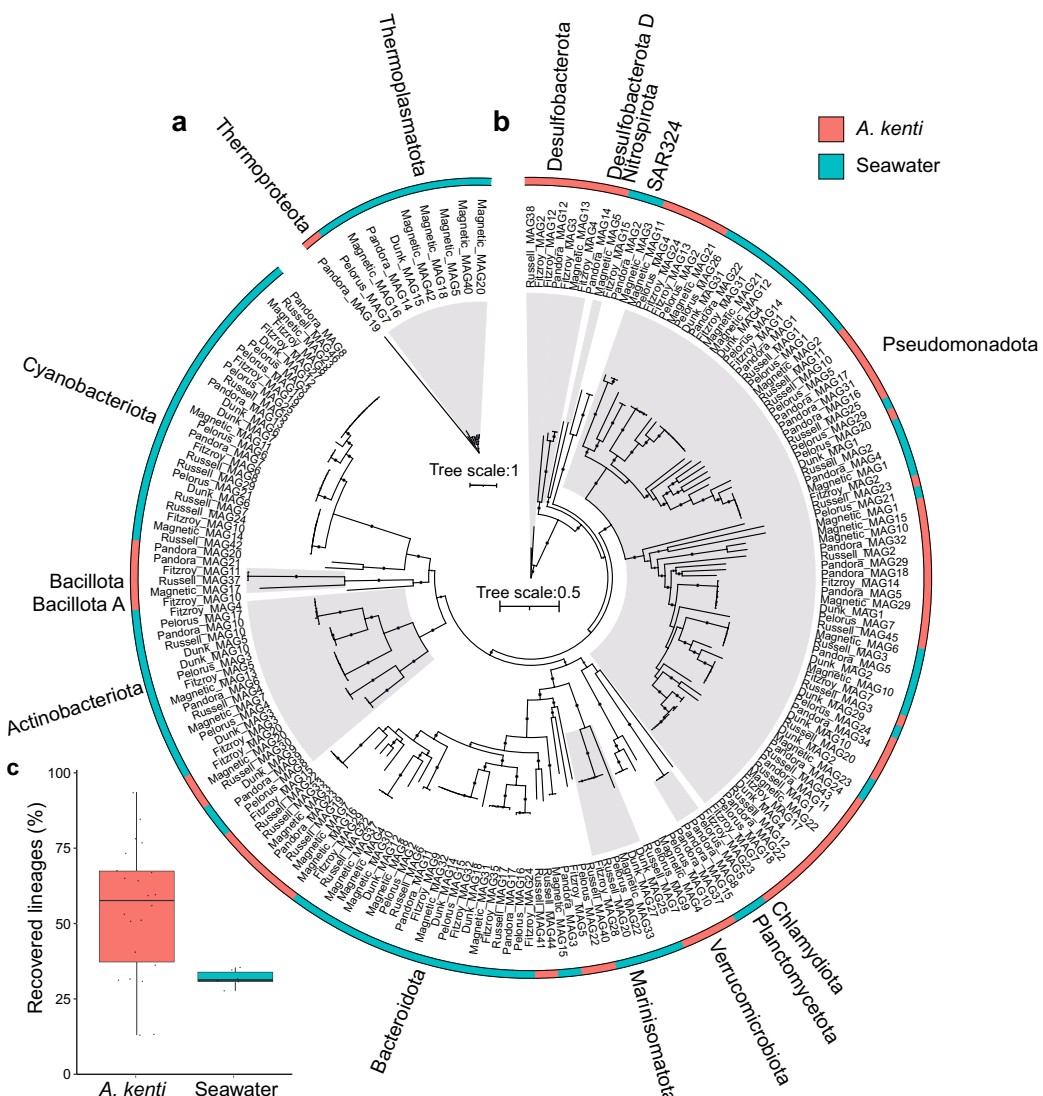

**Fig. 1 | Recovery of metagenome-assembled-genomes (MAGs) from *A. kenti* and adjacent seawater.** Phylogenetic trees were constructed from 102 *A. kenti*-derived and 99 seawater-derived MAGs from the Archaea (**a**) and Bacteria (**b**). The alternating grey and white shading along tree branches in (**a**) and (**b**) denote the boundary of the external phyla labels. The colour strip identifies the sample type the MAGs are specific to, based on their prevalence and normalised relative abundances (see Fig. 2). The proportion of genera captured by these MAGs, based on single copy marker gene appraisal, are shown in (**c**). Box plots display the median, 25th and 75th percentiles, and ±1.5× the inter-quartile range, n = 22 for *A. kenti*, n = 6 for seawater. Source data are provided as a Source Data file and in Supplementary Data 1.

the *A. kenti*-specific MAGs displayed significantly larger genome sizes, lower GC content, greater numbers of predicted genes, and reduced coding density, despite a greater variation relative to those specific to seawater (Fig. 3a−d). This perhaps reflects the breadth of niche types experienced by microorganisms within the more nutrient-rich coral holobiont compared to seawater[5], including stable and specific symbionts in the skeletal matrix and tissues[35], high density coral-associated microbial aggregates (CAMAs) within tissues[34,36], epibiotic and intracellular bacteria associated with the Symbiodiniaceae[37], and a combination of stable and opportunistic symbionts within coral mucus[38] (see Supplementary Note 3). Consistent with the overall genomic differences between the *A. kenti*- and seawater-specific MAGs, significant differences were observed in the presence-absence of CAZy (PERMANOVA, $F = 7.9$, $P$-adj. < 0.01), KO ($F = 11.8$, $P$-adj. < 0.01), and Pfam ($F = 14.3$, $P$-adj. < 0.01) gene annotations (Fig. 3e−g). However, phylogeny was responsible for much of the variation between the recovered MAGs, regardless of their habitat type (Fig. 3e−g).

## Molecular mechanisms indicative of a host-associated lifestyle

Functional enrichment analysis identified 24 CAZy, 1666 KOs, and 1950 Pfams that were significantly enriched in *A. kenti*-specific MAGs, compared with 7 CAZy, 454 KOs, and 467 Pfams that were enriched in the seawater-specific MAGs (Fisher's Exact Test, $P$-adj. ≤ 0.05; Supplementary Data 2; Fig. 4a−c). Exploration of the significantly enriched functional genes indicated adaptations of the *A. kenti* microbiome to the host-associated niche and revealed potential mechanisms underpinning the establishment and maintenance of the *A. kenti*-microbiome symbiosis. For example, genes encoding bacterial chemotaxis and motility (e.g., *che*, *fli*, and *flg* operons, *motA*, and *flaG*), pili and fimbriae (e.g., *pilIGLNUVZ*, *fimTV*), and polysaccharide biosynthesis as part of self-encapsulation (alginate export, PF13372) or biofilm formation (*exoP/vpsO*, *vpsNM*, *epsC*, *pelACEFG*, *gacSA*), were significantly enriched in the *A. kenti*-specific MAGs and may help these bacteria to sense and navigate towards coral chemical cues and establish symbiotic relationships[39,40] (Fisher's Exact Test, $P$-adj. ≤ 0.05; Supplementary Data 2). These significantly enriched motility and biofilm

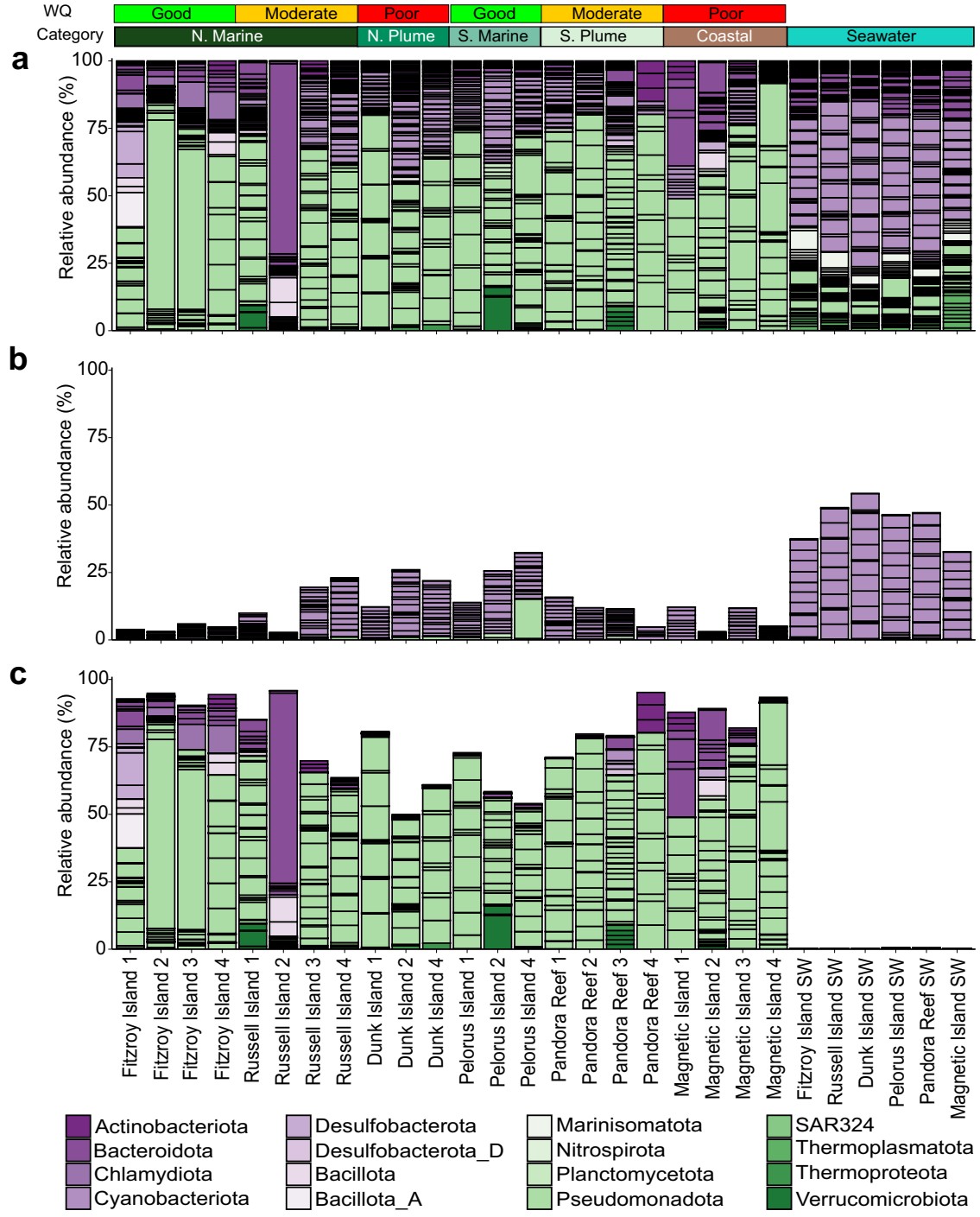

**Fig. 2 | Distribution of 201 MAGs recovered from *A. kenti* and seawater along two water quality gradients of the Great Barrier Reef.** Prevalence and normalised relative abundances (%) of all MAGs (**a**), the 20 MAGs recovered from *A. kenti* that displayed specificity to seawater (**b**), and the 82 *A. kenti*-specific MAGs (**c**). MAGs are coloured by their phylum-level taxonomic assignments and the colour strip represents the two water quality (WQ) gradients sampled in this study and their respective categories. Source data are provided as a Source Data file.

formation genes were present in ~86% (*n* = 71) of the *A. kenti*-specific MAGs, with 12 MAGs classified to the Gammaproteobacteria encoding many (70%–86%) of the identified genes associated with these phenotypes, including *Endozoicomonas* sp. (Russell_MAG12, Fitzroy_-MAG7, Pandora_MAG22, Pelorus_MAG18), *Ferrimonas* sp. (Russell_MAG43), *Thalassotalea* sp. (Pandora_MAG24), and *Alcanivorax* sp. (Dunk_MAG1, Fitzroy_MAG14, Magnetic_MAG29, Pandora_MAG5, Pelorus_MAG7, Russell_MAG45). Although chemotaxis can be employed by coral pathogens[41], the ability to navigate towards a coral host may also provide a competitive advantage for commensal

microorganisms[39]. Indeed, these gammaproteobacterial genera have previously been identified within the *A. kenti* microbiome using 16S rRNA gene sequencing, and/or that of other coral species from the Indo-Pacific[27,31,42], suggesting that phenotypes and behaviours such as chemotaxis, motility, and biofilm formation likely facilitate the success of coral-associated symbionts across hosts.

Following initial host colonisation, proliferation within a coral host may be enhanced by the ability of the microbiome to evade the host's immune responses and modulate host cell metabolism[43,44]. Indeed, several eukaryote-like repeat protein domains (ERPs), such

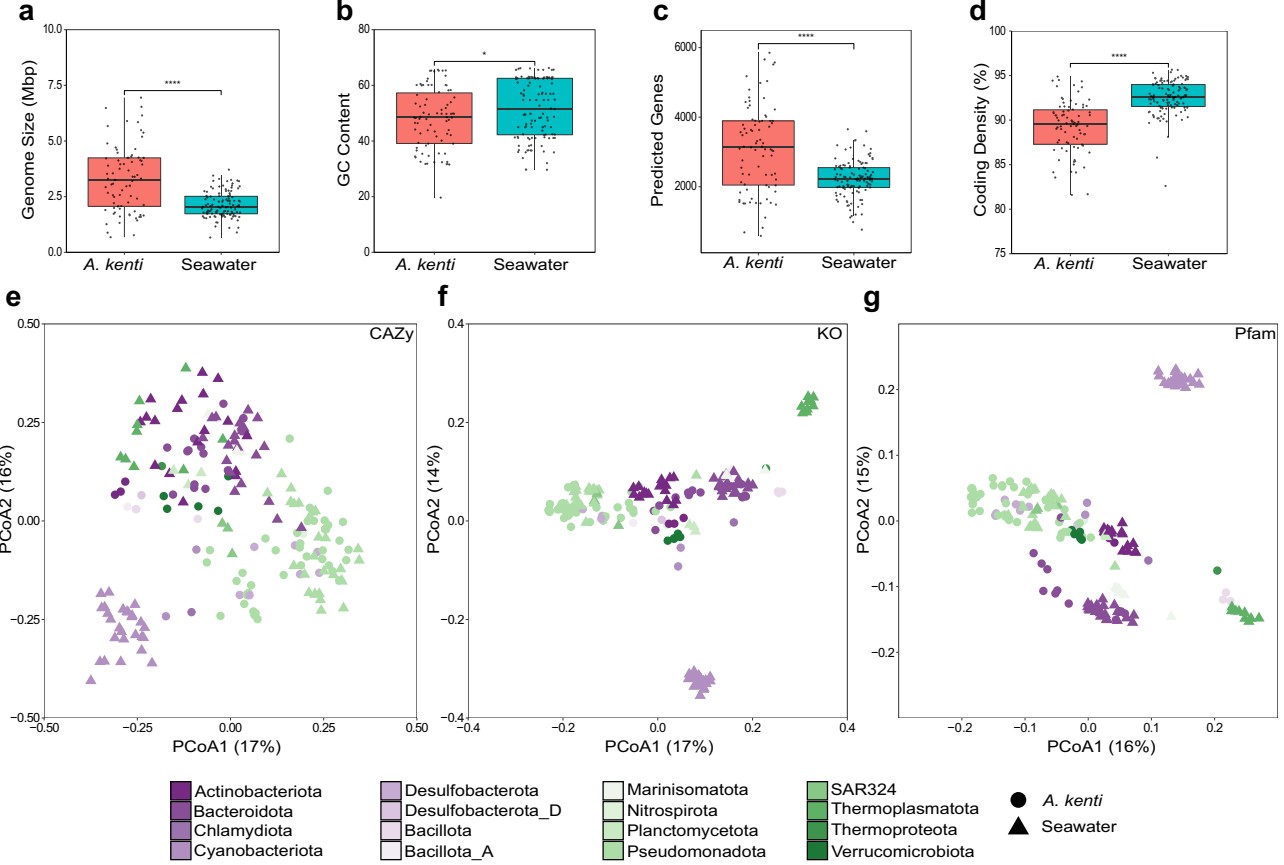

**Fig. 3 | Genomic and functional features of MAGs recovered from *A. kenti* and seawater.** Significant differences were observed between the *A. kenti*- (*n* = 82) and seawater-specific (*n* = 119) MAGs in terms of genome size (**a**), GC content (**b**), the number of predicted genes (**c**), and percent coding density (**d**; asterisks denote significance values derived from two-sided Kruskal Wallis non-parametric tests, indicating FDR adjusted *P* values, where **** represents <0.0001 (a = 5.7 × 10⁻⁹; c = 2.2 × 10⁻⁷; d = 1.6 × 10⁻¹⁶ respectively) and * represents 0.03). Box plots display the median, 25th and 75th percentiles, and ±1.5× the inter-quartile range. Genomic features of MAGs can be found in Supplementary Data 1. Principal coordinates analysis of the annotated functional potential of *A. kenti*- and seawater-specific MAGs based on binary Bray-Curtis dissimilarity matrices of the presence-absence of carbohydrate active enzymes (CAZy; **e**), Kyoto Encyclopedia of Genes and Genomes (KEGG) Orthology (KO; **f**), and protein families (Pfam; **g**).

as ankyrin repeats, a leucine rich repeat, a WD40-like repeat, and tetratricopeptide repeats, involved in immune response evasion in both pathogens and symbionts were enriched in *A. kenti*-specific MAGs relative to those specific to the seawater (Fisher's Exact Test, *P*-adj. ≤ 0.05; Supplementary Data 2; Fig. 4c). In total, 96% (*n* = 79) of *A. kenti*-specific MAGs encoded at least one ERP, with the highest number of annotations of any given ERP (PF12796, Ank_2) observed in three MAGs of the *Endozoicomonas* sp. (Magnetic_MAG22, Fitz-roy_MAG17, Pandora_MAG11) at 85, 78, and 74 copies, respectively. Moreover, protein families implicated in cell-cell interactions in eukaryotes[45–47] were significantly enriched in *A. kenti*-specific MAGs, such as signalling pathways (START domain, PF01852; and phospholipase D, PF00614), and eukaryotic cell attachment, integrity, and stability domains (PF10116; PF00028; PF01875; cardiolipin formation, *clsA_BC*; Fisher's Exact Test, *P*-adj. ≤ 0.05; Supplementary Data 2). In addition, protein families implicated in bacterial virulence, such as intracellular growth, and Type III, IV, and VI secretion systems that function in the injection of effector proteins into host cells[48], were significantly enriched in *A. kenti*-specific MAGs (Fisher's Exact Test, *P*-adj. ≤ 0.05; Supplementary Data 2; Fig. 4c). These secretion systems were encoded by 47% (*n* = 39) of *A. kenti*-specific MAGs of the Bacteroidota, Chlamydiota, Desulfobacterota, Bacillota, Pseudomonadota, and Verrucomicrobiota (Supplementary Data 2). Collectively, such mechanisms may facilitate the establishment of dense areas of microbial growth within CAMAs *in hospite*[36] and may

regulate the intracellular transport of key molecules between members of the holobiont[45]. Indeed, the prevalence of these genes within the *Endozoicomonas* sp. MAGs which are known to predominate CAMAs[36], and other intracellular microorganisms including novel *Mycoplasmataceae*, provide evidence for the specialisation of these lineages within the *A. kenti* holobiont. Taken together these findings indicate that members of the *A. kenti* microbiome may establish and maintain symbioses through evasion of phagocytosis, binding to host cells, and potentially engaging in direct and indirect host cell modification. Elucidation of the exact nature of these mechanisms at the molecular level is required and may provide insights into the resilience or dysbiosis of coral holobionts in the face of anthropogenic stress.

In addition to interactions with coral host cells, genes significantly enriched in *A. kenti*-specific MAGs also indicated dynamic interactions between the microbiome and bacteriophage or foreign DNA. For instance, proteins associated with lysogenic phage, CRISPR-associated proteins (CAS1, 2, 5, and 7), toxin-antitoxin systems, and DNA binding and repair mechanisms were significantly enriched in *A. kenti*-specific MAGs relative to those specific to seawater (Fisher's Exact Test, *P*-adj. ≤ 0.05; Supplementary Data 2; Fig. 4c). Overall, 90% (*n* = 74) of *A. kenti*-specific MAGs across all phyla encoded protein families associated with viral defence mechanisms. In particular, the phage CI repressor protein was present in 30 of the *A. kenti*-specific MAGs spanning 5 phyla,

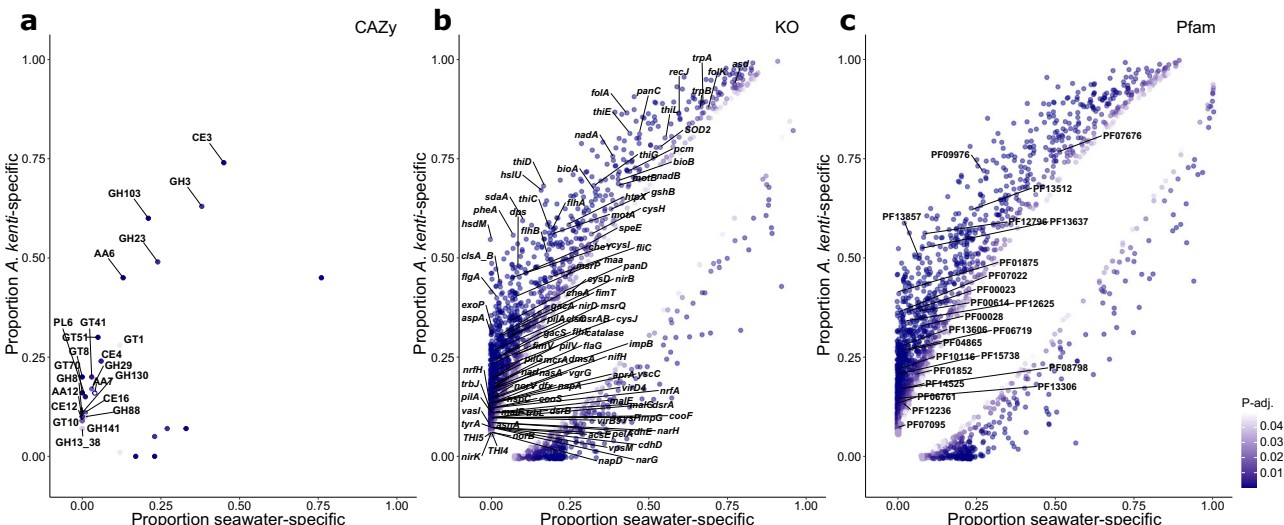

**Fig. 4 | Distribution of significantly enriched functional genes between the *A. kenti*-specific and seawater-specific MAGs.** Significant enrichment was determined using a two-sided Fisher's Exact Test between the *A. kenti*-specific (*n* = 82) and seawater-specific MAGs (*n* = 119) annotated with (**a**) CAZy, (**b**) KO, and (**c**) Pfam databases with correction for multiple testing by controlling the False Discovery Rate. Points are shown as the proportion of total MAGs encoding that functional gene within each category and coloured based on the value of the adjusted P-values. For clarity, only key genes enriched in *A. kenti*-specific MAGs discussed in the main text are labelled. Full details of significantly enriched functional genes are provided in Supplementary Data 2.

suggesting lysogenic phage may be prevalent within this holobiont. This is consistent with previous observations suggesting that lysogeny dominates over viral lysis in high microbial biomass environments such as coral mucus[49]. Previous studies have revealed the extensive diversity of viruses infecting all members of the coral holobiont, and at times contributing to the maintenance and decline of reef health[50–52]. Our results suggest that the *A. kenti* microbiome has a greater requirement for immunity against viruses and foreign DNA compared to microorganisms of the surrounding seawater, similar to previous observations within the sponge holobiont[53]. *A. kenti* MAGs may evade viral infection through a range of molecular mechanisms, facilitating persistence within the holobiont. Interestingly, viral auxiliary metabolic genes implicated in cyanobacterial photosynthesis and encoding putative antioxidants have previously been observed in the *A. kenti* holobiont[54], thus our findings of bacteria-phage interactions as a discriminate feature of the *A. kenti* microbiome suggests this is an important avenue of future research.

In recent years, it has been hypothesised that the microbiome plays a role in the maintenance of coral health[55–57]. Correspondingly, several genes which may support the holobiont during times of stress were significantly enriched within the *A. kenti*-specific MAGs. Among these were genes encoding the antioxidants catalase, glutathione synthase, superoxide reductase and superoxide dismutase, in addition to an oxidative-repair enzyme (*msrA/B, msrPQ*), and a heat shock protein (*hslU*) (Fisher's Exact Test, *P*-adj. ≤ 0.05; Supplementary Data 2). All *A. kenti*-specific MAGs encoded at least one, to a maximum of 6, of these antioxidants, suggesting widespread potential to mitigate damage from reactive oxygen species within the *A. kenti* microbiome. This may reflect the relatively harsh environment experienced by microorganisms associated with corals, which are exposed to daily fluctuations in oxygen and UV-light levels under normal holobiont homeostasis[58]. On the other hand, microbiome encoded antioxidants may help to mitigate cellular damage due to thermal stress, as has previously been shown in cultured Symbiodiniaceae-associated bacteria[11]. Further exploration of the expression and action of the identified antioxidants *in hospite* is required to determine their overall impact on microbiome persistence and in the maintenance of *A. kenti* holobiont health.

## Nutrient cycling contributing to *A. kenti* holobiont function

Several genes significantly enriched in the *A. kenti*-specific MAGs belonged to key nutrient cycling pathways within the holobiont, providing another example of specialisation between the *A. kenti*- and seawater-specific microorganisms. This included the metabolism of polysaccharides and photosynthates common to coral mucus[59,60], such as transcriptional regulators for the uptake and catabolism of arabinose (PF12625, PF14525, PF06719), CAZy families targeting alginate (PL6), mannan (GH130), fucose (GH29, GH141), starch and glycans (GH15, GH13_38), and genes encoding the metabolism and transport of the photosynthates aspartate (*nadB, asnA, asd, aspA, panD, pcm*) and maltose (*maa, malEFG*; Fisher's exact test, *P*-adj. ≤ 0.05; Supplementary Data 2; Fig. 4a–c). These genes were collectively encoded by 99% (*n* = 81) of the *A. kenti*-specific MAGs, with the Bacteroidota, Desulfobacterota, Pseudomonadota, and Verrucomicrobiota encoding the greatest total numbers. One *Draconibacterium* sp. MAG (Russell_MAG41; Prolixibacteraceae, Bacteroidota) appeared to be a specifically adapted to this niche, encoding 73% of the identified genes associated with coral-mucus metabolism. This organism is a facultative anaerobe, suggesting potential adaptation to the oxygen-variable niche of the surface boundary layer to exploit localised polysaccharides and photosynthates[58,61]. Collectively, these results reflect the notion that coral mucus represents a rich source of organic compounds within the oligotrophic reef seascape that is an important driver of holobiont community structure at the coral-seawater interface[62].

Nitrogen availability is considered a primary determinant of Symbiodiniaceae population density and controls the release and translocation of photosynthates to the coral host[63], as such nitrogen fixation is believed to be an important function mediated by the coral microbiome[64]. Key genes encoding nitrogen fixation (*nifHDK* and *nifENB*) were significantly enriched within 11% (*n* = 9) of the *A. kenti* specific-MAGs, of the Desulfobacterota (*n* = 5), Bacteroidota (*n* = 2) and Pseudomonadota (*n* = 2; Fisher's Exact Test, *P*-adj. ≤ 0.05; Supplementary Data 2; Fig. 5). Ammonia derived from nitrogen fixation, or remineralised nitrogen, could provide an energy source for the microbiome through the sequential steps of nitrification[65]. However, none of the MAGs encoded complete nitrification, although the Thermoproteota *JACEMXO1* sp. (Pandora_MAG19) encoded ammonia monooxygenase subunit A (*amoA*) and the novel

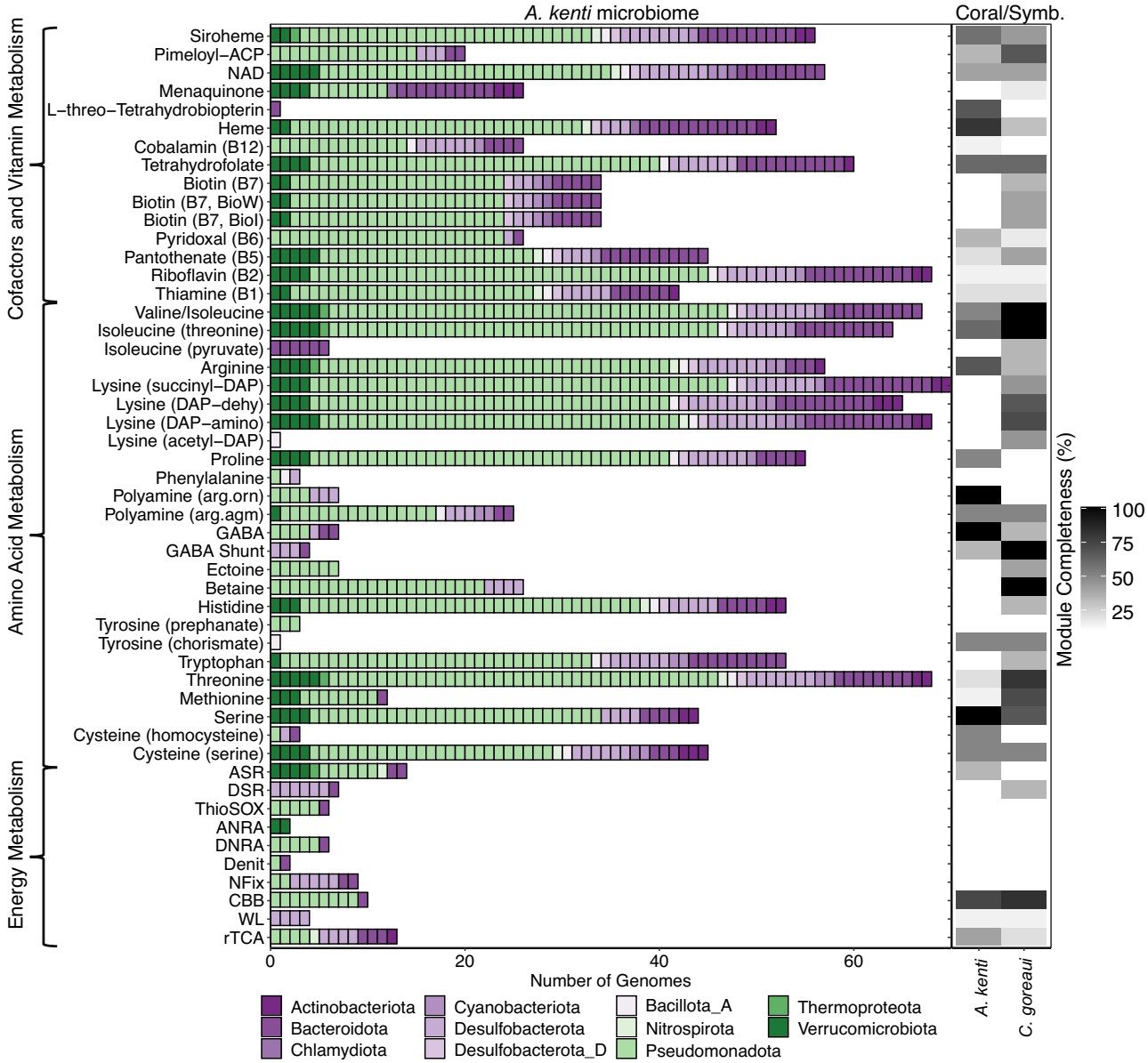

**Fig. 5 | Metabolic overview of key KEGG modules (>75% complete for MAGs) central to energy metabolism, amino acid, and vitamin biosynthesis within the *A. kenti* holobiont.** *A. kenti* and *C. goreaui* annotations were performed on publicly available protein predictions (see Methods) and are represented as KEGG module completeness (%; right side of figure). MAGs are coloured by their phylum-level taxonomic assignments. Source data are provided as a Source Data file.

rTCA reverse tricarboxylic acid cycle, WL Wood-Ljungdahl pathway, CBB Calvin-Benson-Bassham cycle, NFix dinitrogen fixation, Denit denitrification, DNRA dissimilatory nitrate reduction to ammonia, ANRA assimilatory nitrate reduction to ammonia, ThioSOX thiosulfate oxidation, DSR dissimilatory sulfate reduction, ASR assimilatory sulfate reduction.

Nitrospirota *Bin75* sp. (Magnetic_MAG5) encoded nitrite oxidoreductase (*nxrAB*), reflecting previous functional gene studies of coral nitrification[66]. Nitrite can be utilised by other holobiont members for assimilation or energy conservation, with three different nitrite reductases (*nrfAH*, *nirBD*, *nirK*) of the dissimilatory nitrate reduction to ammonia (DNRA) and denitrification pathways significantly enriched in the *A kenti*-specific MAGs (Fisher's Exact Test, *P*-adj. ≤ 0.05; Supplementary Data 2; Fig. 4b). Consistently, nitrate reductases of DNRA, denitrification, and the assimilatory nitrate reduction to ammonia (ANRA) pathways (*napAD, narGHI*, and *nasA*, respectively) were also significantly enriched (Fisher's Exact Test, *P*-adj. ≤ 0.05; Supplementary Data 2; Fig. 4b). Of the MAGs encoding assimilatory pathways, *Nitratireductor aquibiodomus* (Pandora_MAG31; Rhizobiaceae, Pseudomonadota) displayed

prevalence across 60% of *A. kenti* samples at relative abundances between 0.1 and 7.4%. The metabolic potential of this MAG suggests it plays a potential key role in holobiont nitrogen cycling. Although the distribution of several of these lineages across the *A. kenti* colonies was patchy (Fig. 2; see Supplementary Note 4) and functional validation of these largely anaerobic processes within *A. kenti* is required, the genomic potential for anaerobic nitrogen metabolisms is consistent with previous research demonstrating the activity of nitrate reduction within Caribbean scleractinian corals and bacterial isolates from the coral holobiont[10]. The greater number of lineages with complete assimilatory as opposed to reduction pathways (Fig. 5), suggests low nitrogen levels and limited nitrogen loss from the *A. kenti* holobiont, in-line with previous observations of low denitrification rates in corals relative to assimilatory

pathways or nitrogen fixation[65]. Thus the activity of these MAGs may contribute to *A. kenti* holobiont homeostasis, by retaining nitrogen within the system[10,67].

Acroporid corals are autotrophic, and therefore primarily dependent upon carbon derived from Symbiodiniaceae photosynthesis[28]. Here, four autotrophic carbon fixation pathways were also identified within the *A. kenti*-specific MAGs spanning 16 families from 5 different phyla (see Supplementary Note 5). These included the reverse tricarboxylic acid cycle, the modified hydroxypropionate-hydroxybutyrate cycle, the Calvin-Benson-Bassham cycle, and the Wood-Ljungdahl (WL) pathway, suggesting prokaryotic carbon fixation may supplement the holobiont carbon pool (Fig. 5), as has recently been suggested for the *P. lutea* holobiont[18]. Of these, genes encoding key steps within the WL pathway (*cooSF, cdhED*, and *acsE*) were significantly enriched within the *A. kenti*-specific MAGs compared to those specific to seawater (Fisher's Exact Test, *P*-adj. ≤ 0.05; Supplementary Data 2). The complete WL pathway was restricted to MAGs of the *Desulfobacter* sp. (Fitzroy_MAG3, Fitzroy_MAG4, Magnetic_MAG13) and *Desulforhopalus* sp. (Pandora_MAG12), all of which encoded the key WL enzyme carbon monoxide dehydrogenase. It is notable that 6 of the *A. kenti*-specific MAGs, including *Desulfobacter* sp. and *Desulforhopalus* sp., in addition to *Chlorobium_A marina* and *Ferrimonas* sp., encoded two autotrophic carbon fixation pathways. This is unusual in both symbiotic and free-living bacteria and has been suggested as a mechanism to increase the efficiency of carbon fixation, for example within sulfur-oxidising gammaproteobacterial symbionts of marine tubeworms[68]. Redundancy in autotrophic carbon fixation pathways may provide a functional advantage if fixed carbon becomes limiting during times of stress[67]. In the future, the expression of these pathways and quantification of their rates in situ are required to validate the importance of microbiome contributions to the *A. kenti* carbon pool.

Sulfate is unlikely to be a limiting nutrient within the coral holobiont, as it is readily available within marine waters and can be assimilated by the host and Symbiodiniaceae[69], and is often utilised by marine microorganisms as a source of sulfur through its assimilatory reduction to sulfide[70]. Nevertheless, genes encoding extracellular sulfate transport (*cysPUWA*), and key steps within assimilatory sulfate reduction (ASR; e.g., *cysD, cysH, cysJI*) and dissimilatory sulfate reduction pathways (DSR; e.g., *aprAB, dsrAB*) were significantly enriched in the *A. kenti*-specific MAGs relative to those specific to seawater (Fisher's Exact Test, *P*-adj. ≤ 0.05; Supplementary Data 2). This suggests that differences in sulfur utilisation or availability may occur between the *A. kenti* and seawater niches. As an example, methylated sulfur compounds may represent preferential sulfur sources *in hospite*[71] as the metabolism of reduced sulfur compounds is less energetically expensive[72]. Indeed, *Acropora* sp. are known to contain high concentrations of methylated sulfur compounds, such as dimethylsulfoniopropionate (DMSP) which is produced by both the animal and Symbiodiniaceae[73,74]. Within the genes significantly enriched in *A. kenti*-specific MAGs were those encoding the anaerobic dimethyl sulfoxide (DMSO) reductase enzyme (*dmsABC*; Fisher's Exact Test, *P*-adj. ≤ 0.05; Supplementary Data 2), which catalyses the reduction of DMSO to the climate active gas dimethyl sulfide (DMS). Moreover, phylogenetic placement of genes involved in DMSP lysis via the cleavage (*ddd* genes) and demethylation (*dmd* genes) pathways identified putatively functional *dddD* (*n* = 14), *dddL* (*n* = 3), *dddP* (*n* = 6), and *dddW* (*n* = 8) genes in 28 *A. kenti*-specific MAGs of the Pseudomonadota (*n* = 27) and Desulfobacterota (*n* = 1; Supplementary Data 3). Notably, the *dddD* gene, which converts DMSP to DMS, acetate, and 3-hydroxypropionate[75], was present in MAGs recovered independently from each sampling location. All 9 *Endozoicomonas* sp. MAGs encoded *dddD*, extending recent reports of the potential for DMSP metabolism by cultured strains of *Endozoicomonas acroporea*[9]. DMSP demethylation to methanethiol was encoded by 10 MAGs of the Gamma- (*n* = 6) and Alphaproteobacteria (*n* = 4), including representative MAGs from

each sampling site (Supplementary Data 3). Three MAGs encoded genes for both DMSP cleavage and demethylation, including Gammaproteobacteria belonging to the *QNFE01* family (Magnetic_MAG15) and *Luminiphilus* sp012270045 (Pandora_MAG34), and *CABZJG01* sp. within the *Rhodobacteraceae* (Pelorus_MAG20). Interestingly, this suggests these taxa may be able to switch between the two pathways in a concentration dependent manner[76,77], or in response to changing environmental conditions[78]. Functional validation of these genes is required to truly define the role of these microorganisms in modulating the production of DMS within the *A. kenti* holobiont.

Scleractinian corals display distinct patterns of essential and non-essential amino acid metabolism[79,80]. Functional annotation of *A. kenti* and *C. goreaui* proteins demonstrated incomplete pathways for a range of amino acids, including histidine, tryptophan, and lysine, in addition to vitamins, such as thiamine, riboflavin, biotin, and cobalamin (Fig. 5). Instead, these pathways were widespread across the *A. kenti*-specific MAGs, spanning a diversity of different phyla (Fig. 5). Moreover, significantly enriched functional genes within the *A. kenti*-specific MAGs included those encoding the biosynthesis of the polyamine spermidine (*nspC* and *speE*), and the amino acids tyrosine/phenylalanine (*pheA, tyrA*), tryptophan (*trpAB*), and serine (*sdaAB*; Fisher's Exact Test, *P*-adj. ≤ 0.05; Supplementary Data 2). In addition, genes encoding five B-vitamins, namely thiamine (B$_1$; *thiCDEGHLM, THI45*), nicotinate (B$_3$; *nadABCD, pncB, ushA, yrfG*), pantothenate (B$_5$; e.g., *panCD*), biotin (B$_7$; *bioABMN*), and folate (B$_9$; e.g., *folAK*) were also enriched in the *A. kenti*-specifc MAGs (Fisher's Exact Test, *P*-adj. ≤ 0.05; Supplementary Data 2). These findings suggest that amino acid and B-vitamin metabolism are an important functional trait of the *A. kenti* microbiome that may be essential to support the nutrient requirements of the *A. kenti* holobiont. This is highlighted by the absence and incomplete nature of these pathways in *A. kenti* and *C. goreaui* (Fig. 5), and across several other coral host and Symbiodiniaceae genomes[18,81,82].

## Latitudinal heterogeneity in the *A. kenti* microbiome underpinned by water quality

Significant differences in microbial community composition, and the associations between environmental variables underpinning community-level and functional gene dissimilarity, were explored using constrained ordination and functional gene enrichment. Prior to statistical analyses, additional dereplication was performed to group closely related MAGs recovered independently from the different sites (see Methods). This resulted in 63 *A. kenti*-specific and 49 seawater-specific MAGs, which displayed significant differences in community composition between them, and across the inshore reefs (*P* = 0.001, *F* = 6.06, *n* = 28; Goodness of fit, *r*² = 0.78, *P* = 0.001; Table S1). Up to ~80% of the variation in MAG community structure was attributed to the separation between the seawater and *A. kenti* samples, followed by the latitudinal influence of the northern (Dunk Island, Russell Island, and Fitzroy Island) and southern (Magnetic Island, Pandora Reef, and Pelorus Island) sampling gradients (spanning poor to improved water quality, respectively; Fig. 6a; Supplementary Fig. 2). Significant differences in *A. kenti* community composition were associated with the water quality categories of these sites (*P* = 0.001, *F* = 2.3, *n* = 22; Goodness of fit, *r*² = 0.68, *P* = 0.001; Table S2), primarily reflecting the differences between the North Marine (Fitzroy and Russell Islands) and degraded Coastal (Magnetic Island) reefs (Fig. 6b; see Methods). Post-hoc testing identified the North Marine and Coastal *A. kenti* microbiomes as being significantly distinct to each other (Table S2). While the North Marine communities were also significantly distinct to the degraded North Plume (Dunk Island) and less-impacted South Marine (Pelorus Island) samples (*P*-adj. < 0.05). However, the differences between the North Plume, South Marine, South Plume (Pandora Reef), and Coastal communities were not significant (Table S2), displaying similarities in their composition that suggests an interplay between

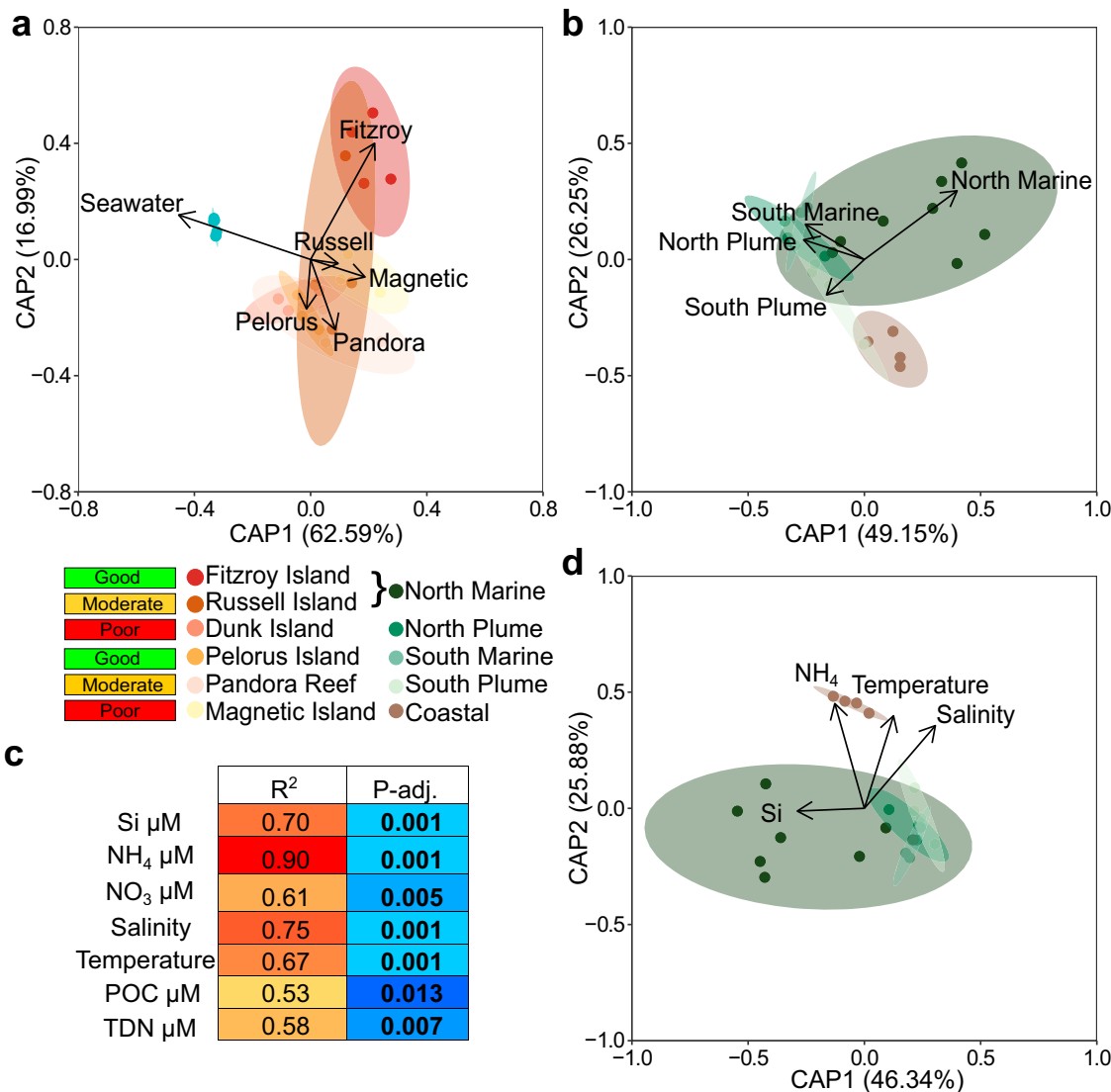

**Fig. 6 | Constrained ordination of MAG-based microbial community composition reveals latitudinal heterogeneity driven by water quality.** Dissimilarity between seawater and *A. kenti* community composition based on the Hellinger-transformed Bray-Curtis distance metric of overall dereplicated MAGs (*n* = 112), analysed using principal coordinates analysis with the Island site of origin of samples (*n* = 28) as a factor (**a**), and water quality categories of samples (*n* = 22) as a factor (**b**). Model significance was tested using one-way analysis of variance. Significant water quality parameters contributing to *A. kenti* community dissimilarity were identified using factor fitting to an ordination (**c**), and the structuring nature of the strongest correlates across the water quality categories was visualised using constrained ordination (**d**). In (**b**–**d**), the ordinations and significance tests were performed on *A. kenti*-specific MAGs (*n* = 63) and samples (*n* = 22) only. Model outputs are provided in Supplementary Data 5–8. POC particulate organic carbon; TDN total dissolved nitrogen. *P* values were corrected for multiple testing made by controlling the False Discovery Rate.

latitude and water quality driving *A. kenti* microbial community dissimilarity.

Differences in coral and seawater microbiome community structure are often related to geographic distances and reef water quality[83–85]. The water quality parameters significantly correlated to the heterogeneity in *A. kenti* microbiome community composition encompassed physical, chemical, and biological variables (Fig. 6c, d). Subsequent ordination with the four variables displaying the strongest correlation ($R^2 > 0.65$, *P*-adj. = 0.001) revealed that these variables explained ~72% of the variation in the first two axes, with Si associated with the separation of the North Marine samples, and $NH_4$, temperature, and salinity underpinning the partitioning of the Coastal samples from the other categories (Fig. 6d). Corresponding with the structuring nature of temperature and dissolved inorganic nitrogen ($NH_4$ and $NO_3$), MAG-based gene clusters significantly enriched between the water quality categories included

those identified as heat-shock proteins, molecular chaperones, and nitrate reductases (Mann Whitney U, $P < 0.05$; Supplementary Data 4). The prevalence of clusters associated with reducing or repairing protein mutations and misfolding (e.g., cluster_84722, cluster_55609, cluster_45348, cluster_11927) and periplasmic nitrate reductase subunit *napA* (cluster_54900), specifically in the Coastal category, suggests adaptation of the microbiome to increased temperature stress and nutrients at this site. This is consistent with the typically warmer, relatively more nitrogen-rich waters of the inshore GBR, which may be more directly impacted by anthropogenic activity[86]. The ability to overcome protein damage to maintain important phenotypes is also indicative of resilience in the face of environmental perturbation[87], which may suggest one mechanism of *A. kenti* holobiont adaptation within the typically more stressful coastal inshore GBR. These findings build upon recent observations of acquired resilience in *A. kenti* recruits

through parental environmental history[22,23], suggesting a role for the microbiome in environmental adaptation.

In summary, 82 medium-high quality MAGs specific to the *A. kenti* microbiome revealed molecular mechanisms putatively underpinning the functioning of the *A. kenti* holobiont. This unprecedented number of MAGs from coral tissues provided phylogenetic and genomic context to important metabolic pathways, connecting *A. kenti* microbial community structure to holobiont function. Adaptations to life in the coral host niche included behavioural and phenotypic processes, such as chemotaxis, motility, and biofilm formation, that may be driven by host and Symbiodiniaceae metabolites. In addition, key functions, such as carbon and nitrogen fixation and the metabolism of essential histidine, lysine, tryptophan, and B-vitamins, may supplement the activity of *A. kenti* or the holobiont. Most of these functions were distributed across diverse prokaryotic taxa, with little evidence for functional specialisation except for autotrophic carbon and nitrogen transformations, which were restricted to specific lineages. The widespread invertebrate associated *Endozoicomonas* sp. appear to be important symbionts of *A. kenti*, displaying extensive metabolic flexibility, including the genomic capacity to metabolise DMSP. Further, the *A. kenti* microbiome encoded several pathways of host evasion and manipulation, highlighting a range of mechanisms that may be integral to symbiosis establishment or maintenance.

Perhaps reflecting the combination of obligate, facultative, and transient symbionts that associate with corals, and the challenges in separating holobiont compartments and organisms, many of the recovered MAGs displayed patchy distributions across the *A. kenti* colonies. However, significant differences in *A. kenti* microbiome community composition were observed across the inshore GBR, with the most pronounced of these between the marine and coastal reef sites from the northern and southern ends of the two sampling gradients, respectively. Temperature and dissolved inorganic nitrogen were among the most significant environmental variables driving this dissimilarity, with significant enrichment in functional gene clusters encoding heat-shock proteins, molecular chaperones, and nitrate reductases, providing insight into phenotypic adaptations underlying these statistical relationships. Collectively, this study provided community-level understanding of the functioning of the *A. kenti* microbiome and highlights the mechanisms and microorganisms which may support this holobionts' health in the face of natural environmental variability and anthropogenic stress in the future.

## Methods

All sample collections were performed in accordance with the Great Barrier Reef Marine Park Authority (GBRMPA) regulations, under permit number G14/36802.1.

### Experimental design and sample collection

*A. kenti* colonies (note: revision from *A. tenuis*[20]) were collected along two gradients of increasing water quality from inshore reefs during a voyage of the RV *Cape Ferguson* in the tropical wet season (20th−23rd February 2015; permit G14/36802.1), following the route of Cooke and colleagues[24]. The northern gradient, spanning poor to improved water quality, included Dunk Island, Russell Island, and Fitzroy Island, respectively (Supplementary Fig. 2). The southern sampling gradient encompassed the sites Magnetic Island, Pandora Reef, and Pelorus Island, from poor to improved water quality. These sites can also be broadly classified as marine (Fitzroy, Russell, and Pelorus Islands), river-plume impacted (Dunk Island and Pandora Reef), and coastal (Magnetic Island)[24,88]. At each site, branches (from the actively growing apical to intermediate branches) from four *A. kenti* colonies were fragmented (~10 × 10 cm²) and thoroughly rinsed with sterile calcium and magnesium-free artificial seawater (CMFSW) [0.45 M NaCl, 10 mM KCl, 7 mM $Na_2SO_4$, 0.5 mM $NaHCO_3$] to limit carry-over of microorganisms not specifically associated with the coral holobiont. One

sample (5 L) from the surrounding seawater was collected from each site and 5 μm pre-filtered onto a 0.2 μm Sterivex filter (Merck Millipore; #SVGP01015) for community comparisons to the coral microbiome.

The environmental conditions at each site were determined according to the GBR Marine Monitoring Programme using standard protocols to determine temperature, salinity, and concentrations of particulate organic carbon (POC), particulate phosphorus (PP), dissolved inorganic phosphorus (DIP), total dissolved phosphorus (TDP), particulate nitrogen (PN), dissolved inorganic nitrogen ($NO_3^-$, $NO_2^-$, $NH_4^+$), total dissolved nitrogen (TDN), silica (Si), and chlorophyll α (Chl-α)[89].

### Microbial enrichment, nucleic acid extraction, and metagenomic sequencing

*A. kenti* tissues were removed from the skeleton using a sterile air pick and 30 ml sterile CMFSW. Blastate was homogenised using a sterile glass dounce by vertically rotating the pestle for 30 s. The methodology described by Robbins and colleagues[18] was used to enrich microbial cells from homogenised coral holobiont samples, with some modifications. Briefly, two enzyme treatments were applied to break down coral mucus and tissue structures, including (i) α-amylase (Sigma; 2 μ/ml, 37 °C, 15 min at 150 rpm) followed by (ii) Dispase II (Gibco; 10 mg/ml, 37 °C, 15 min at 150 rpm), with stocks prepared in sterile ultrapure water. Sequential filtration through sterile 200 and 100 μm mesh followed by a slow speed centrifugation (300 × *g*, 15 min) at room temperature was used to remove intact tissues, skeletal debris, and Symbiodiniaceae cells. A sterile 8.0 μm Isopore membrane filter (Millipore) was used to remove remaining Symbiodiniaceae and coral cells prior to centrifugation at 10,000 × *g* to pellet microorganisms (10 min, room temperature). Microorganisms were preserved in filter-sterilised (0.1 μm) glycerol-TE[90] 2X PBS buffer and snap-frozen in liquid nitrogen prior to storage at −80 °C.

Microbial pellets were thawed on ice and filtered through a sterile 5 μm Isopore membrane filter (Merck Millipore; #TMTP02500) to remove residual Symbiodiniaceae, alongside a negative control comprised of 1 ml ultrapure water, immediately prior to DNA extraction using the MoBio Ultra Clean Microbial DNA Isolation kit (now QIAGEN, DNeasy Ultraclean Microbial kit; #10196-4). DNA was purified using the Zymo Genomic DNA Clean and Concentrator kit (Zymo Research; #D4010) and quantified using the Qubit High Sensitivity dsDNA kit (Invitrogen; #Q32851) prior to library preparation. Metagenomic libraries (2 × 150 bp) were prepared with the Illumina Nextera XT library preparation kit (#FC-131-1096) following the manufacturer's protocol, or using a low-input protocol[91] depending on the DNA concentration, which included 12 or 20 cycles of PCR respectively. An additional negative control comprised of ultrapure water was included for the library preparation. Libraries were visualised on the TapeStation using a Genomic DNA Tape (Agilent; #5067-5365), and only those with a discernible peak of the expected size and quantifiable DNA concentration were selected for sequencing (n = 22/24 *A. kenti* samples, 6/6 seawater samples and 1/2 negative control samples from DNA extraction and library preparation). All *A. kenti* samples underwent shallow sequencing (~2 Gbp) on the Illumina HiSeq2500 platform (Ramaciotti Centre for Genomics, University of New South Wales). After basic inspection of data quality, all samples were sequenced to a greater depth (~20 Gbp; including the seawater samples) and one *A. kenti* sample from each sampling location underwent deep sequencing (~80−100 Gbp per sample). As samples were sequenced multiple times from the same starting library, reads were concatenated prior to downstream analysis.

### Genome-resolved metagenomics workflow

Metagenomic reads mapping to the publicly available reference genomes of *A. kenti*[24] (http://aten.reefgenomics.org/), *Cladocopium C15*[18]

(http://plut.reefgenomics.org/cladocopium_download/), and *Cladocopium goreaui*[26] (https://doi.org/10.14264/uql.2019.745), and those representing PCR duplicates, were removed prior to downstream processing (see Supplementary Methods). In total, between 56 and 90% of metagenomic reads were removed through this rigorous QC process, with most of these reads mapping to the coral genome (Supplementary Fig. 1). The remaining reads were assembled using the metaspades.py script of Spades (v3.13.0)[92], setting the -m flag to a maximum of 1500 for the deep metagenomes. Quality controlled reads were mapped to the resulting scaffolds using CoverM 'make' with default settings (v0.2.0-alpha7; https://github.com/wwood/CoverM), and the generated BAM files and scaffolds were used as input to the custom ensemble binning tool UniteM (v1.0.0; https://github.com/dparks1134/UniteM/), using the commands 'bin', 'profile', and 'consensus' to produce a non-redundant set of MAGs from the binning methods: GroopM2 (https://github.com/timbalam/GroopM)[93], MaxBin using both the 40 and 107 marker sets[94], and MetaBAT2 and MetaBAT1 using the settings --verysensitive --sensitive --specific --veryspecific --superspecific[95]. Contigs identified as outliers based on GC content and tetranucleotide distance were removed from the MAGs using RefineM with the commands 'scaffold_stats', 'outliers', and 'filter_bins' (v0.0.24; https://github.com/dparks1134/RefineM;[96]) with default settings. The quality (defined as completeness - 3 × contamination[18]) of the resulting MAGs was determined using CheckM (v1.1.2[97]);

### MAG taxonomy and phylogenetic tree construction

MAGs were assigned taxonomy based on the standardised taxonomic ranks of the Genome Taxonomy Database (GTDB, r202[98]) using the accompanying tool GTDB-tk (v1.5.0[99]). To reduce redundancy between *A. kenti* biological replicates, but not to eliminate beta-diversity, MAGs with a quality score ≥ 50 were dereplicated based on the sampling site they were recovered from using dRep (v2.5.4[100]; default settings), including the additional quality filtering step of only retaining MAGs ≥75% complete. As only one seawater sample was collected per site, the seawater MAGs were dereplicated across the six seawater samples using dRep as above. The extent of the microbial communities encompassed by the dereplicated MAGs was determined using SingleM 'appraise' (v0.12.1), by comparing single copy ribosomal proteins clustered at the genus level, following[101], in the MAGs and metagenomic reads (https://github.com/wwood/singlem). The final dataset of refined dereplicated MAGs was used to create phylogenetic trees. Briefly, the multiple-sequence-alignments for the bacterial and archaeal MAGs generated separately by GTDB-tk, were used as input to IQ-TREE[102] with the LG + G model for Bacteria, and the LG + C10 + F + G model for Archaea, and the flags -nt 10 -b 1000. The phylogenetic trees were visualised using the Interactive Tree of Life (iTOL[103]) and decorated with GTDB taxonomy and other metadata (Supplementary Data 1).

### Functional annotation

EnrichM 'annotate' (v0.4.15; https://github.com/geronimp/enrichM) was used to annotate predicted proteins of all MAGs, and the publicly available predicted proteins of *A. kenti* and *C. goreaui*, with carbohydrate active enzymes (CAZy), Kyoto Encyclopedia of Genes and Genomes (KEGG) Orthology (KOs), and protein families (PFams). EnrichM 'classify' (v0.4.15) was used to determine the completeness of KEGG Modules (%) based on KO annotations. To identify functional motifs in specific genes of interest (*dddX*, *dmdA*, cuMMO, *nxrAB*, *nifH*, *rbcL*, *aclA*), GraftM (v0.13.1; https://data.ace.uq.edu.au/public/graftm/7/[104]); was used to search the predicted proteins of MAGs and place then into a phylogenetic tree, using publicly available GraftM packages. To confirm the GraftM results, genes identified as being from selected functional clades were subject to a BLASTP search against the NCBI nr database to confirm their identify compared to known functional motifs (accessed 25/03/2021).

### Microbial community composition and MAG specificity

Mean coverages were calculated by mapping the QC'd reads to the dereplicated MAGs using CoverM (v0.4.0) and the flags -m mean --min-read-percent-identity 0.95, --min-read-aligned-percent 0.75 and retaining the 'min-covered-fraction' default of 10%. The resulting table was normalised to account for differences in metagenome sequencing depth between samples, by scaling to the smallest library size (Gb; number of reads × average read length) and transformed to relative abundance. MAG specificity was determined according to the relative abundance and prevalence of each MAG across the *A. kenti* and seawater samples. This was determined following three *a posteriori* criteria: an *A. kenti* MAG should have a greater mean relative abundance in *A. kenti* compared to seawater samples (*A. kenti*: seawater relative abundance ratio >1), should display <50% prevalence within the seawater, and represent <0.1% relative abundance in each seawater sample. In satisfying these three criteria, we allow for the hypothesis that some coral-associated symbionts are horizontally acquired from the surrounding seawater but will display greater prevalence and higher relative abundances within the host.

### Statistical analyses

Genome characteristics including genome size, GC content, number of predicted genes, and coding density, were compared between *A. kenti*- and seawater-specific MAGs using a Kruskal Wallis non-parametric test (R, v4.1.1). Permutational analysis of variance (PERMANOVA) was used to determine significant differences in the presence-absence of functional genes based on CAZy, KO, and Pfam annotations with a binary Bray Curtis dissimilarity matrix generated using vegdist for input into adonis2, as implemented by vegan (v2.5.7)[105]. Principal coordinates analysis was used to visualise the dissimilarity between the presence-absence of functional gene annotations, and specific differences in functional gene content were determined using Fisher's Exact Test as employed by EnrichM's 'enrichment' function (Supplementary Data 2).

Closely related MAGs representative of strains (ANI ≥ 99%) recovered independently from different sites were removed by additional dereplication (as above) prior to biogeography analyses. Mean coverages were calculated for the 'overall' dereplicated MAGs by mapping the QC'd reads using CoverM (v0.4.0) and transformed to relative abundance after normalising for differences in metagenome size (as above). Relative abundances were Hellinger transformed and a Bray-Curtis distance matrix generated using the vegdist function of vegan (v2.5.7). A model was built from the dissimilarity matrix using either the 'Island Origin' or 'Water Quality' categories as a factor with principal coordinates analysis, employing the capscale function in vegan (v2.5.7). Model significance was tested using analysis of variance with 'goodness of fit' of the factor levels determined using envfit. Post-hoc pairwise tests to identify which factors differed significantly were performed using the pairwise.factorfit function of RVAideMemoire (v0.9-80) with 999 permutations. Ordinations were plotted using ggord (v1.1.6) with additional modifications performed through the tidyverse (v1.3.1) implementation of ggplot2 (v.3.3.5)[106] and the ggarrange function of ggpubr (v0.4.0)[107].

Water quality metadata were Z-score standardised using the scale function in R (v4.1.1), and any collinear variables (Pearson $R > 0.7$ or $< -0.7$ as determined by the cor function in R[108]); were removed. The standardised water quality parameters were fitted to the *A. kenti* water quality category ordination as vectors using envfit, with *P* values corrected for multiple testing made by controlling for the False Discovery Rate (denoted by '*P*-adj.'). Significant variables with an $R^2 > 0.65$ were then selected to build a third ordination using the *A. kenti* sample matrix as input to the capscale function of vegan (v2.5.7).

EnrichM (v0.4.15) 'annotate' was used to parse MMSeqs2[109] to produce orthologous clusters reflecting all gene content from the overall dereplicated MAGs across the sampling gradient. Significant

enrichment of gene clusters was determined using a Mann Whitney U Test through EnrichM 'enrichment'. The normalised relative abundances of the overall dereplicated MAGs were used to weight cluster frequencies by the relative abundance of the organisms encoding them, using water quality categories as factors for the Mann Whitney U Test ($P < 0.05$; Supplementary Data 4).

## Reporting summary

Further information on research design is available in the Nature Portfolio Reporting Summary linked to this article.

## Data availability

The sequence data generated in this study have been submitted to NCBI under BioProject PRJNA545004, with BioSample accessions spanning SAMN37503415-SAMN37503442 [https://www.ncbi.nlm.nih.gov/biosample/?term=SAMN37503415]. Environmental water quality data obtained according to the Great Barrier Reef Marine Monitoring Programme are publicly available at https://www.aims.gov.au/data. The *A. kenti* genome used in this study is available at http://aten.reefgenomics.org/, *Cladocopium* C15 sp. at http://plut.reefgenomics.org/cladocopium_download/, and *Cladocopium goreaui* https://doi.org/10.14264/uql.2019.745. The *A. kenti* and seawater MAGs metadata (i.e., taxonomy, quality, genomic features) generated in this study are provided in the Supplementary Data files. The Genome Taxonomy Database is publicly available at https://gtdb.ecogenomic.org/downloads. GraftM packages used in this study are available at https://data.ace.uq.edu.au/public/graftm/7/. Source data are provided with this paper.

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

## Acknowledgements

We would like to thank the captain and crew of the R/V *Cape Ferguson* in addition to Johnston Davidson, Paul Costello, Dr Russell Carpenter, and Dr Kathy Morrow, for assistance with sampling. We are grateful to Serene Low from the Australian Centre for Ecogenomics for assistance with library preparation. We thank Dr Jean-Baptiste Raina and Dr Kaylyn Tousignant for critically reading a draft of this manuscript. This research was funded by the Queensland Government DSITIA Accelerate Partnerships award to the University of Queensland on behalf of the Australian Institute of Marine Science (AIMS), the Australian National University, Bioplatforms Australia, the Great Barrier Reef Foundation, the Great Barrier Reef Marine Park Authority, and James Cook University (2014; awarded to G.W.T. and D.G.B.), and the Australian Research Council Discovery Project Scheme (Grant DP160103811 awarded to G.W.T. and D.G.B.).

## Author contributions

D.G.B. and G.W.T. conceived and designed the study and acquired funding. S.C.B. and D.G.B. collected the samples and performed the microbial enrichment protocol. L.F.M. and S.J.R. performed DNA extractions. L.F.M. and G.W.T. devised the bioinformatic workflow, and L.F.M. performed the bioinformatic analyses, interpreted the data, produced the figures and tables, and wrote the manuscript. M.C. contributed to the biogeography analyses. S.J.M. and G.W.T. critically revised the manuscript. All authors approved the final manuscript.

## Competing interests

The authors declare no competing interests.
