## [Peer Review File · Nature Communications]

A genome-centric view of the role of the *Acropora kenti* microbiome in coral health and resilienceREVIEWER COMMENTS

Reviewer #1 (Remarks to the Author):

I find this study by Messer et al. very interesting, and a valuable contribution to the field. Many words have been written about coral-microbe symbiosis over the past couple decades, but very few studies have produced data such as this. The generation of enough high quality microbial genomes from enough host samples to conduct functional comparisons across habitats marks a substantial step forward in our ability to create and refine hypotheses about the role of microbes in the coral holobiont. In particular, in contrast to most metagenomic approaches, I appreciate the increased ability to focus on microbial taxa that are more meaningfully associated with the coral, as opposed to ‘functional’ analyses of isolated genes that often derive environmental contaminant organisms, or those with superficial interactions with the host. I also appreciate that the authors are careful with their language and often highlight that even with these data, functional roles are still hypothetical and need to be verified with more targeted experimentation.

I have a number of comments below; mostly superficial and relating to clarity of language. Overall, the authors’ conclusions are generally well supported, or are otherwise given the appropriate caveats (I point out a couple of minor exceptions, below). The most substantial doubt I have is with regard to the classification of ‘A. kenti-specific’ taxa. What is the probability that some of these taxa are more common in coral samples than in water not because they are meaningfully interacting with living coral tissue (part of the ‘holobiont’ if the term is to be useful), but because they are commensally associated with dead coral skeleton, or detritus that has settled or been eaten by polyps (/primarily/ heterotrophic doesn’t mean /completely/), or are transferred from other organisms that are in contact with the coral, like turf algae or epibiotic crustaceans? Some discussion of these possibilities would be useful. Further, because the water samples were collected and processed in a manner that is distinct from the coral samples, it is possible that some taxa that were only found in the coral samples are there not because they live in association with the animals, but because they are introduced or highlighted during sample processing – for instance via a contaminated air pick or contaminated coral-DNA-extraction-specific reagents. I think it is reasonable to proceed with the assumption that this is not the case, but the authors may be able to do a better job discussing this in the manuscript. Lines 65-96 of the supplementary

methods are a good inclusion, but (1) a brief mention of the existence of this control would be useful in the main text; (2) more information could be included about what a 'DNA extraction and library preparation negative control' is. If this is a sample of air pick solution that has gone through all the steps that the coral samples went through, then great! If not, then at what point was this control included? After the microbial enrichment protocol; only at the MoBio DNA kit step? What is the likelihood that all the previous steps introduced or preferentially isolated some of the taxa analyzed?; (3) if this was truly a single control sample, how representative is it of the probability of contamination or bias? If the Cuticobacterium MAGs that were appropriately highlighted as suspicious are truly contaminants, doesn't that have further implications for the other MAGs that weren't found in the control? I will emphasize, however, that the authors seem to have already put more effort into addressing these kinds of caveats than is standard in the field, so I do not want these criticisms to be /too/ penalizing. I think the procedure and discussion are reasonable overall.

Minor details:

Line 23: The first sentence in the manuscript is awkward. Grammatically, maybe all it needs is a period before the however and a comma after it. But this was distracting at the start.

Lines 50-53: Whether a 'holobiont' is an 'ecological unit' is far from a settled question, and phylogenetic surveys cannot establish whether this is the case. The holobiont concept is an invention that may or may not help understand patterns – it is not a discovery that has been 'revealed'. There is likely a more precise way to convey what you want, here.

Line 93: 'holobionts' needs a possessive apostrophe.

Lines 98-101: This sentence is a bit hard to parse. Consider rewording it for clarity.

Line 105: 'river-plume impacted (i.e. degraded)'. This may actually be an important aspect to address at this and a couple other points in the paper. Are the 'degraded' sites truly 'degraded'? I.e., these sites are different because they are impacted by a river plume. But is this temporally distinct from how they were in the past? Are this study's conclusions truly generalizable to 'degraded' reef sites, or are we just comparing two different habitat types; each of which have some degree of similarity and difference to how they have existed for millenia?

Line 125: I just thought it was interesting to highlight here that Thermoproteota used to be Thaumarchaeota. You could highlight the same for numerous other taxa, and also – why not

use the updated names for 'Pseudomonadota' and 'Thermodesulfobacteriota' instead of 'Proteobacteria' and 'Desulfobacterota'? Just semantics, but I found it a little inconsistent

Line 148: What is sample-specific? Aren't there always going to be slight differences in abundance between samples just due to noise? Was there some kind of within-sample replication that I missed that allowed for a determination of 'significant' differences in abundance between samples?

Lines 161-165: This is an awkward list, putatively related to the previous sentence, masquerading as its own sentence?

Line 241: Maybe 'This is consistent with previous observations'

Line 254: I would be more circumspect with this statement. The role of the microbiome in coral health is still mostly hypothetical. Especially the role of /natural variation/ in the microbiome, which is what these statements generally imply. (Showing that bacterial compositions change in response to differences in coral health has the causation backward, and even solid experimental evidence that bacteria can affect coral health does not solidly establish that changing epibiotic bacterial composition via environmental differences has any meaningful effect...)

Lines 310-313: It seems a bit of a stretch to call something with 60% prevalence, including samples with 0.1% abundance, a microbe with a 'key role in holobiont...' Given those data, it is at best facultative, and at worst, a passerby who simply has the potential to be helpful.

Line 346: Sulfate is unlikely to be /a/ limiting nutrient?

Line 363: I think there's an extra comma in there after 'pathways'. Hard to track some of these sentences with all the parentheticals.

Line 422 (and elsewhere): Although it's a common way to talk about multivariate associations, saying parameters 'drive' heterogeneity is fairly strongly attributing a causal interpretation where there should be a simple description of pattern. It is possible (in many cases probable) that these parameters are causally related to microbiome composition, but it is also possible that they are themselves 'driven' by a lurking variable (in this case probably geographic) that itself directly causes the differences in the microbiome. I would advise against this casual use of causal language.

Line 458: I wouldn't say these data 'revealed' Endozoicomonas as 'important symbionts'. It is reasonable to hypothesize that these taxa are important, but it really is still just a hypothesis. They could still turn out to be opportunistic commensals (or some strains are,

and others are important to their host...)

Line 530: GroopM doesn't seem to be available at this link

Line 571-573: This is a bit hard to parse. Abundance ratio between A kenti: seawater > 1? I think I get it, but could be written a bit clearer.

Thank you all,

Ryan McMinds

Reviewer #2 (Remarks to the Author):

The authors present a comprehensive analysis of metagenome assembled genomes derived from *Acropora kenti* corals and nearby seawater from locations in the Great Barrier Reef. Through the analysis, they examine the functional potential of the coral microbiome compared to that of the surrounding seawater, host and algal symbiont genomes, and also explore changes across natural gradients in water quality.

Overall, I enjoyed reading the paper. The authors were very clear in the hypotheses relating to microbial functional roles and environmental gradients. While the paper is quite dense, the authors do a nice job of synthesizing the relevant information and placing the results in context as we go by using the Results & Discussion section. I believe this paper builds nicely on the growing body of literature from some of the same authors, in which they generate and explore the functional potential of MAGs from demosponges and *Porites lutea*. I think the coral reef microbial ecology community will be broadly interested in this paper for both validating and spurring further experimental research in reef microbial ecology.

I have no major comments or suggestions for restructuring the paper or analyses conducted. My one broad suggestion is to finely go through the methods and ensure all parameters and packages used are included and additionally include scripts and analytical code used for the analysis. With so many bioinformatic and statistical analyses outlined in the paper, reproducibility of research can be a major challenge. Most of my suggestions below are minor and include some suggestions of places where I think more methodological

detail could be included.

- Introduction L105-108. I found the description of the two locations, water quality, and latitude confusing. Was the gradient of higher to lower water quality across 2 degrees of latitude? Were the two locations 2 degrees apart and they individually had water quality gradients? Instead of “i.e. degraded”, could you say “i.e. poor water quality”? Perhaps including a supplementary graph of the sampling locations, water quality gradient, and river-plume would be useful here. Rather than calling them plume, marine, and coastal, you could refer to them as low, medium, and high-impact?

- Several (at least 4-5 sections) in the results & discussion refer to Figure S2. Is there any reason it is not in the main text since it highlights the genes that are distinct between seawater and *A. kenti* microbiomes? I feel like it would be useful to have in the main text to compare the two sets of MAGs.

- L317-318 – I love the section on different aspects of nitrogen cycling in corals. I was hoping to see more discussion of the Babbitt 2021 paper (currently citation #10). The paper conducted nitrogen cycling rate measurements in Caribbean corals and coral-derived cultures of nitrogen cycling microorganisms. In particular they measured nitrate reduction (among other pathways). While this was done in corals in a different geographic region, it warrants further discussion in this section, and directly addresses the statement “functional validation of these largely anaerobic processes is required”.

- L442 “suggesting a role for the microbiome in environmental adaptation” – The section on microbiome changes corresponding to water quality is interesting, and it may also have to do with the geographic location of the corals. It might be worth mentioning that geographic differences, which are often related to water quality, have been shown to impact the structure of the coral microbiome in previous studies using 16S rRNA gene sequencing (Williams et al 2022, <https://peerj.com/articles/13574/>; Becker et al 2023, <https://academic.oup.com/pnasnexus/article/2/9/pgad287/7259987>)

- L519-520 – The citations might be off. For example, citation 86 does not seem appropriate for the *Cladocodium* C15 genome. Additionally, for ease of reproducibility, I suggest you include the repository accession or link to the publicly available genomes. Sometimes it can be difficult to identify the location from a citation alone.

- L526 – What were your CoverM parameters? Default?

- L528 – For using UniteM, are the parameters the “—verysensitive”, etc.?
- L533 – For RefineM, were there specific cutoffs you used for the outliers, or was it default?
- L555 – Did you use a specific command within EnrichM? Which one? Any specific parameters?
- L555-556 – Did you generate the annotated predicted proteins of MAGs, A. kenti, and C. goreau? If so, how?
- L594 – Did you use a specific method for normalizing for different metagenome sizes? If so, what method?
- L598 and rest of paragraph – Can you provide the capscale models (and envfit) you used?
- Methods general – will you provide scripts used in a repository or supplementary figure for publication?
- L617 – which statistical tests?
- Data availability – Does this BioProject include both raw sequence reads as well as the MAGs? Will the analytical code be available too?
- L924 – “Metabolic overview of key modules” – How were these identified? What tool? Was this in the methods?
- L948 – Supplementary Table Captions – When I downloaded the supplementary Excel files, they had no clear titles. In the top of the Excel sheets or in the tab, could you clearly label, for example “Table S2”. If this could be done for all the supplementary details that would be helpful since I believe they get removed from the file names.
- Figure 2 – The Good to Poor water quality gradient seems misleading. Is it truly a gradient, or was it Good water quality at “S. marine”, medium water quality at “S. plume” and poor at “Coastal”. Similarly, were “N. Plume” and “Coastal” equally poor? Maybe three distinct colors there would be useful.
- Figure 3 – Could you label E, F, and G with the different annotation databases?
- Figure 3 – The gradient color scheme for taxa is somewhat hard to distinguish. Is there a reason it was chosen? Perhaps something more distinct would be useful unless it isn’t critical to the story.
- Figure 4 – Will you provide definitions for the energy metabolism acronyms?
- Figure 5 – For consistency, could you also label the different plumes by their water quality gradient designation?
- Supplementary Information:

- L49 – Any specific parameters for Seqpurge?
- L50 – It was unclear from the Robbins paper citation for *Cladocopium* C15 where to get the genome, though I did ultimately track it to reefgenomics.org? Maybe also cite that repository directly? If there is an accession number for it, add that too.
- L50 – Same for *Cladocopium* goreau. The citation had a clear location, but the repository also has a citation that you can include: Chen, Yibi, González-Pech, Raúl A., Stephens, Timothy G., Bhattacharya, Debashish, and Chan, Cheong Xin(2019). Revised genome sequences and annotations of six Symbiodiniaceae taxa. The University of Queensland. Data Collection.<https://doi.org/10.14264/uql.2019.745>
- L54-64 – I was confused why the reads were assembled with megahit, when in the body of the text it said that Spades was used (L524). Can you clarify what went on here?

Response to reviewer's for:

'A genome-centric view of the role of the *Acropora kenti* microbiome in coral health and resilience'

REVIEWER COMMENTS

N.B. Reviewer comments are shown in black font, with our point-by-point response displayed immediately below in bold red typeface.

Reviewer #1 (Remarks to the Author):

I find this study by Messer et al. very interesting, and a valuable contribution to the field. Many words have been written about coral-microbe symbiosis over the past couple decades, but very few studies have produced data such as this. The generation of enough high-quality microbial genomes from enough host samples to conduct functional comparisons across habitats marks a substantial step forward in our ability to create and refine hypotheses about the role of microbes in the coral holobiont. In particular, in contrast to most metagenomic approaches, I appreciate the increased ability to focus on microbial taxa that are more meaningfully associated with the coral, as opposed to 'functional' analyses of isolated genes that often derive environmental contaminant organisms, or those with superficial interactions with the host. I also appreciate that the authors are careful with their language and often highlight that even with these data, functional roles are still hypothetical and need to be verified with more targeted experimentation.

We are delighted that the reviewer believes our manuscript represents a substantial step forward for the coral microbiome field. We have addressed the reviewer's specific comments below, indicating where the changes have been made in the revised manuscript.

I have a number of comments below; mostly superficial and relating to clarity of language. Overall, the authors' conclusions are generally well supported, or are otherwise given the appropriate caveats (I point out a couple of minor exceptions, below). The most substantial doubt I have is with regard to the classification of 'A. kenti-specific' taxa. What is the probability that some of these taxa are more common in coral samples than in water not because they are meaningfully interacting with living coral tissue (part of the 'holobiont' if the term is to be useful), but because they are commensally associated with dead coral skeleton, or detritus that has settled or been eaten by polyps (/primarily/ heterotrophic doesn't mean /completely/), or are transferred from other organisms that are in contact with the coral, like turf algae or epibiotic crustaceans? Some discussion of these possibilities would be useful.

The classification of '*A. kenti*-specific' metagenome-assembled genomes (MAGs) that were more abundant and more prevalent in the *A. kenti* samples compared to seawater, was to focus the metabolic reconstructions on the MAGs most likely to be interacting with the living host. To ensure that the microorganisms recovered were derived from living tissues, the fragments used for microbiome enrichment were collected from the top of the apical branches to the intermediate branches of the basal colony, encompassing the actively growing extensions of the coral animal. Further, upon collection, the coral fragments were thoroughly rinsed with sterile artificial seawater to remove any settled detritus, residual surrounding seawater, or the presence of any cells from other microorganisms previously in contact with the coral surface.

We have added further details to the revised manuscript on lines 493-497 regarding the steps taken in the field to ensure the samples were representative of the *A. kenti* microbiome. We now state, "*At each site, branches (from the actively growing apical to intermediate branches) from four A. kenti colonies were fragmented (~10 × 10 cm²) and thoroughly rinsed with sterile calcium and magnesium-free artificial seawater (CMFSW) [0.45 M NaCl, 10 mM KCl, 7 mM Na₂SO₄, 0.5 mM NaHCO₃] to limit carry-over of microorganisms not specifically associated with the coral holobiont.*".

Further, because the water samples were collected and processed in a manner that is distinct from the coral samples, it is possible that some taxa that were only found in the coral samples are there not because they live in association with the animals, but because they are introduced or highlighted during sample processing – for instance via a contaminated air pick or contaminated coral-DNA-extraction-specific reagents. I think it is reasonable to proceed with the assumption that this is not the case, but the authors may be able to do a better job discussing this in the manuscript. Lines 65-96 of the supplementary methods are a good inclusion, but (1) a brief mention of the existence of this control would be useful in the main text; (2) more information could be included about what a 'DNA extraction and library preparation negative control' is. If this is a sample of air pick solution that has gone through all the steps that the coral samples went through, then great! If not, then at what point was this control included? After the microbial enrichment protocol; only at the MoBio DNA kit step? What is the likelihood that all the previous steps introduced or preferentially isolated some of the taxa analyzed?; (3) if this was truly a single control sample, how representative is it of the probability of contamination or bias? If the *Cuticobacterium* MAGs that were appropriately highlighted as suspicious are truly contaminants, doesn't that have further implications for the other MAGs that weren't found in the control? I will emphasize, however, that the authors seem to have already put more effort into addressing these kinds of caveats than is standard in the field, so I do not want these criticisms to be /too/ penalizing. I think the procedure and discussion are reasonable overall.

As pointed out by the Reviewer, we endeavoured to minimise the risks of contamination during sample processing and metagenomic sequencing and confirmed that none of the MAGs were present in the sequenced control sample. The sequenced negative control of ultrapure water was included at the point of the 5 µm filtration of the microbial enrichment protocol (when the samples were received at the University of Queensland) and was subsequently carried forward through DNA extraction, DNA concentration and

cleaning, and library preparation. All equipment and reagents were prepared under sterile conditions, including use of sterile air-picks, sterile media, sterile glassware and plasticware during the microbiome enrichment procedure. Due to these best efforts, we are confident in the data presented herein.

We have updated our description of the microbiome enrichment procedure to unequivocally state that only pre-sterilised equipment and reagents were used during sample processing, so that lines 503-516 now read, “*A. kenti* tissues were removed from the skeleton using a sterile air pick and 30 ml sterile CMFSW. Blastate was homogenised using a sterile glass dounce by vertically rotating the pestle for 30s. The methodology described by Robbins and colleagues¹⁸ was used to enrich microbial cells from homogenised coral holobiont samples, with some modifications. Briefly, two enzyme treatments were applied to break down coral mucus and tissue structures, including (i) α -amylase (Sigma; 2U/ml, 37°C, 15 min at 150 rpm) followed by (ii) Dispase II (Gibco; 10mg/ml, 37°C, 15 min at 150 rpm), with stocks prepared in sterile ultrapure water. Sequential filtration through sterile 200 and 100 μ m mesh followed by a slow speed centrifugation (300 \times g, 15 min) at room temperature was used to remove intact tissues, skeletal debris, and Symbiodiniaceae cells. A sterile 8.0 μ m Isopore membrane filter (Millipore) was used to remove remaining Symbiodiniaceae and coral cells prior to centrifugation at 10,000 \times g to pellet microorganisms (10 min, room temperature). Microorganisms were preserved in filter-sterilised (0.1 μ m) glycerol-TE⁹⁰ 2X PBS buffer and snap-frozen in liquid nitrogen prior to storage at -80 °C.”.

Moreover, we have added the description of the controls included in the laboratory processing in the Supplementary Methods on lines 28-40, where we now state, “*In the laboratory, microbial pellets were thawed on ice and filtered through a sterile 5 μ m Isopore membrane filter (Millipore) to remove residual Symbiodiniaceae, alongside a negative control comprised of 1ml ultrapure water, immediately prior to DNA extraction using the MoBio Ultra Clean Microbial DNA Isolation kit. DNA was purified using the Zymo Genomic DNA Clean and Concentrator kit and quantified using the Qubit High Sensitivity dsDNA kit (Invitrogen) prior to library preparation. Metagenomic libraries (2 \times 150 bp) were prepared with the Illumina Nextera XT library preparation kit following the manufacturer's protocol or using a low-input protocol² depending on the DNA concentration, which included 12 or 20 cycles of PCR respectively. An additional negative control comprised of ultrapure water was included for the library preparation. Libraries were visualised on the TapeStation using a Genomic DNA Tape, and only those with a discernible peak of the expected size and quantifiable DNA concentration were selected for sequencing (n=22/24 *A. kenti* samples, 6/6 seawater samples and 1/2 negative control samples from DNA extraction and library preparation).*”. The description of the controls has also been added to the main text of the manuscript on lines 516-518, stating “*Samples were filtered through a sterile 5 μ m Isopore membrane filter (Millipore) to remove residual Symbiodiniaceae, alongside a negative control comprised of 1 ml ultrapure water...*”, and on lines 525-526, “*An additional negative control comprised of ultrapure water was included for the library preparation.*”.

The reviewer points out that some bias may occur in the analysed taxa due to the sample processing. Indeed, the microbiome enrichment procedure was designed to preferentially isolate and concentrate microorganisms associated with coral tissues and mucus for metagenomic sequencing, as without this step, contamination from coral host DNA is overwhelming and allows only rudimentary functional analysis. Our analyses will be biased towards microorganisms that can be liberated from their interactions with coral cells/matrices through the air blasting, homogenisation, enzyme treatment, and size fractionation. The recovery of putative intracellular symbionts, including *Simkania* sp., Mycoplasmataceae, and non-photosynthetic Cyanobacteriota of the order Gastranaerophilales, as well as cosmopolitan coral-associated lineages such as *Endozoicomonas* sp., indicates this was effective. This procedure has been extensively optimised previously by our team (citation 18 in the manuscript) and ensures that the resulting data are robust by reducing host genome contamination to allow greater access to the microbiome for genome-resolved metabolic reconstruction. To make it clear to the reader, we emphasise that this is an enrichment procedure on lines 116-118, where we now state, “*A combination of microbiome-enrichment, for the concentration of microorganisms associated with coral tissues and mucus, and metagenomics (118 Gbp post-QC, see Methods; Figure S1) was used to recover an unprecedented number of coral-associated MAGs from A. kenti of the inshore GBR (n = 22).*”.

Minor details:

Line 23: The first sentence in the manuscript is awkward. Grammatically, maybe all it needs is a period before the however and a comma after it. But this was distracting at the start.

This change has been made.

Lines 50-53: Whether a ‘holobiont’ is an ‘ecological unit’ is far from a settled question, and phylogenetic surveys cannot establish whether this is the case. The holobiont concept is an invention that may or may not help understand patterns – it is not a discovery that has been ‘revealed’. There is likely a more precise way to convey what you want, here.

We agree with the point raised here and have rephrased the sentence accordingly. These lines (50-53) now state, “*Extensive phylogenetic surveys of corals over the last ~20 years have identified hundreds to thousands of other distinct microorganisms, including bacteria, archaea, fungi, protists, and their viruses, which collectively represent the coral holobiont* ^{5,6}.”.

Line 93: ‘holobionts’ needs a possessive apostrophe.

This change has been made.

Lines 98-101: This sentence is a bit hard to parse. Consider rewording it for clarity.

This sentence has been rephrased for clarity. These lines (97-100) now state, “*Recently, genome-wide analyses of A. kenti and Cladocypium have revealed their*

genetic diversity across the GBR^{24,29}. A. kenti colonies appear as distinct populations at coastal sites along the inshore GBR yet maintain consistency in their dominant Cladocopium sp. across different locations²⁴.”.

Line 105: ‘river-plume impacted (i.e. degraded)’. This may actually be an important aspect to address at this and a couple other points in the paper. Are the ‘degraded’ sites truly ‘degraded’? I.e., these sites are different because they are impacted by a river plume. But is this temporally distinct from how they were in the past? Are this study’s conclusions truly generalizable to ‘degraded’ reef sites, or are we just comparing two different habitat types; each of which have some degree of similarity and difference to how they have existed for millenia?

The reviewer makes a valid point and we have replaced ‘degraded’ with ‘poor water quality’ as what we really are referring to here are the physicochemical environmental differences experienced by corals at the reefs sampled. This was informed by the decadal-scale water quality monitoring of the Great Barrier Reef carried out by the Australian Institute of Marine Science as part of the Reef 2050 Plan Marine Monitoring Program (cited in reference 30). We have updated our description of the sampling sites so that these lines (103-108) now state, “Here, visually healthy colonies from six locations on the GBR, which spanned 2° of latitude from the northernmost to southernmost site, were sampled for genome-resolved metagenomics to elucidate microbial functional roles. The variability in environmental water quality between these sites, which included two (one northern and one southern) gradients of river-plume impacted (i.e. poor water quality) to less impacted reefs^{24,30}, provided a natural laboratory to explore the factors influencing A. kenti microbial community structure and metabolic potential.”.

Line 125: I just thought it was interesting to highlight here that Thermoproteota used to be Thaumarchaeota. You could highlight the same for numerous other taxa, and also – why not use the updated names for ‘Pseudomonadota’ and ‘Thermodesulfobacteriota’ instead of ‘Proteobacteria’ and ‘Desulfobacterota’? Just semantics, but I found it a little inconsistent.

We have updated the taxonomy throughout the manuscript and Supplementary Notes to be in line with the updated Phyla names of the Genome Taxonomy Database.

Line 148: What is sample-specific? Aren’t there always going to be slight differences in abundance between samples just due to noise? Was there some kind of within-sample replication that I missed that allowed for a determination of ‘significant’ differences in abundance between samples?

We have removed the reference to ‘sample-specific’ as it was confusing, we did not include within-sample replication as the reviewer rightly points out.

Lines 161-165: This is an awkward list, putatively related to the previous sentence, masquerading as its own sentence?

We have improved the phrasing here so that this sentence (lines 161-166) now reads, “*This perhaps reflects the breadth of niche types experienced by microorganisms within the more nutrient-rich coral holobiont compared to seawater ⁵, including stable and specific symbionts in the skeletal matrix and tissue ³⁵, high density coral-associated microbial aggregates (CAMAs) within tissues ^{34,36}, epibiotic and intracellular bacteria associated with the Symbiodiniaceae ³⁷, and a combination of stable and opportunistic symbionts within coral mucus ³⁸.”.*

Line 241: Maybe ‘This is consistent with previous observations’

This change has been made.

Line 254: I would be more circumspect with this statement. The role of the microbiome in coral health is still mostly hypothetical. Especially the role of /natural variation/ in the microbiome, which is what these statements generally imply. (Showing that bacterial compositions change in response to differences in coral health has the causation backward, and even solid experimental evidence that bacteria can affect coral health does not solidly establish that changing epibiotic bacterial composition via environmental differences has any meaningful effect...).

We have taken this comment on board and have rephrased this sentence (lines 257-258) so that it now reads, “In recent years, it has been hypothesised that the microbiome plays a role in the maintenance of coral health ⁵⁵⁻⁵⁷...”.

Lines 310-313: It seems a bit of a stretch to call something with 60% prevalence, including samples with 0.1% abundance, a microbe with a ‘key role in holobiont...’ Given those data, it is at best facultative, and at worst, a passerby who simply has the potential to be helpful.

We agree with this sentiment, as it is the functions encoded by this microorganism, rather than how commonly it is found, that indicates significance for nitrogen cycling. We have rephrased this sentence, so that lines 313-316 now read, “*Of the MAGs encoding assimilatory pathways, Nitratireductor aquibiodomus (Pandora_MAG31; Rhizobiaceae, Pseudomonadota) displayed prevalence across 60% of A. kenti samples at relative abundances between 0.1-7.4%. The metabolic potential of this MAG suggests it plays a potential key role in holobiont nitrogen cycling.*”.

Line 346: Sulfate is unlikely to be /a/ limiting nutrient?

This change has been made.

Line 363: I think there’s an extra comma in there after ‘pathways’. Hard to track some of these sentences with all the parentheses.

The comma has been removed from the sentence, and we noticed the word ‘genes’ was missing and have corrected this.

Line 422 (and elsewhere): Although it's a common way to talk about multivariate associations, saying parameters 'drive' heterogeneity is fairly strongly attributing a causal interpretation where there should be a simple description of pattern. It is possible (in many cases probable) that these parameters are causally related to microbiome composition, but it is also possible that they are themselves 'driven' by a lurking variable (in this case probably geographic) that itself directly causes the differences in the microbiome. I would advise against this casual use of causal language.

We have revised our language throughout the section “*Latitudinal heterogeneity in the A. kenti* microbiome underpinned by water quality”, to remove the suggestion that the correlations are representative of causal relationships. For instance, on lines 411-413 we now state, “Up to ~80% of the variation in MAG community structure was attributed to the separation between the seawater and *A. kenti* samples”; on lines 416-417 we state, “Significant differences in *A. kenti* community composition were associated with the water quality categories of these sites”; on line 428 we state, “The water quality parameters significantly correlated to the heterogeneity in *A. kenti* microbiome community composition encompassed...”; and on lines 432-434 we now state, “...with *Si* associated with the separation of the North Marine samples, and NH_4 , temperature, and salinity underpinning the partitioning of...”.

Line 458: I wouldn't say these data 'revealed' *Endozoicomonas* as 'important symbionts'. It is reasonable to hypothesize that these taxa are important, but it really is still just a hypothesis. They could still turn out to be opportunistic commensals (or some strains are, and others are important to their host...).

We agree with this sentiment and have rephrased this sentence to now read, “The widespread invertebrate associated *Endozoicomonas* sp. appear to be important symbionts of *A. kenti*...”.

Line 530: GroopM doesn't seem to be available at this link

We apologise for this error and have updated the manuscript with the correct link on line 543.

Line 571-573: This is a bit hard to parse. Abundance ratio between *A. kenti*: seawater > 1? I think I get it, but could be written a bit clearer.

We have clarified this description of *A. kenti*-specific MAGs, so that on lines 588-593 we now state, “MAG specificity was determined according to the relative abundance and prevalence of each MAG across the *A. kenti* and seawater samples. This was determined following three a posteriori criteria: an *A. kenti* MAG should have a greater mean relative abundance in *A. kenti* compared to seawater samples (*A. kenti*: seawater relative abundance ratio > 1), should display < 50% prevalence within the seawater and represent < 0.1% relative abundance in each seawater sample.”.

Reviewer #2 (Remarks to the Author):

The authors present a comprehensive analysis of metagenome assembled genomes derived from *Acropora kenti* corals and nearby seawater from locations in the Great Barrier Reef. Through the analysis, they examine the functional potential of the coral microbiome compared to that of the surrounding seawater, host and algal symbiont genomes, and also explore changes across natural gradients in water quality.

Overall, I enjoyed reading the paper. The authors were very clear in the hypotheses relating to microbial functional roles and environmental gradients. While the paper is quite dense, the authors do a nice job of synthesizing the relevant information and placing the results in context as we go by using the Results & Discussion section. I believe this paper builds nicely on the growing body of literature from some of the same authors, in which they generate and explore the functional potential of MAGs from demosponges and *Porites lutea*. I think the coral reef microbial ecology community will be broadly interested in this paper for both validating and spurring further experimental research in reef microbial ecology.

We would like to thank the reviewer for their appraisal of our work.

I have no major comments or suggestions for restructuring the paper or analyses conducted. My one broad suggestion is to finely go through the methods and ensure all parameters and packages used are included and additionally include scripts and analytical code used for the analysis. With so many bioinformatic and statistical analyses outlined in the paper, reproducibility of research can be a major challenge. Most of my suggestions below are minor and include some suggestions of places where I think more methodological detail could be included.

We have added any missing parameters used throughout the Methods and Supplementary Methods to enhance the reproducibility of our analyses (more details are provided in the specific responses below).

- Introduction L105-108. I found the description of the two locations, water quality, and latitude confusing. Was the gradient of higher to lower water quality across 2 degrees of latitude? Were the two locations 2 degrees apart and they individually had water quality gradients? Instead of "i.e. degraded", could you say "i.e. poor water quality"? Perhaps including a supplementary graph of the sampling locations, water quality gradient, and river-plume would be useful here. Rather than calling them plume, marine, and coastal, you could refer to them as low, medium, and high-impact?

We apologise that this description was unclear. The six sites collectively spanned ~220km, representing 2° of latitude between the northernmost to southernmost reefs sampled. We have replaced 'degraded' with 'poor water quality', as this is indeed what we were referring to here. This was informed by the decadal-scale water quality monitoring carried out by the Australian Institute of Marine Science (manuscript reference 30). We have clarified our description of the sites in the Introduction, so that on lines 103-108 we state, "*Here, visually healthy colonies from six locations on the GBR, which spanned 2° of latitude from the northernmost to southernmost site, were*

*sampled for genome-resolved metagenomics to elucidate microbial functional roles. The variability in environmental water quality between these sites, which included two (one northern and one southern) gradients of river-plume impacted (i.e. poor water quality) to less impacted reefs ^{24,30}, provided a natural laboratory to explore the factors influencing *A. kenti* microbial community structure and metabolic potential.”.*

Moreover, we have adopted the reviewer’s suggestion to remove the colour gradient from Figure 2 (see later specific point) and use the ‘Good’, ‘Moderate’, and ‘Poor’ terminology of the inshore reef Marine Monitoring Program. The use of the categories ‘plume’, ‘marine’, and ‘coastal’ is to retain the general contextual information about the sites, which experience different physicochemical characteristics due to their geographic position. Further, these classifications allow continuity with previous studies from these sites, such as that by Cooke and colleagues (citation 24 in the manuscript) which performed the genome analysis of *A. kenti* from some of the same locations as this study. We have added a map showing the six reef sites in what is now Figure S2.

- Several (at least 4-5 sections) in the results & discussion refer to Figure S2. Is there any reason it is not in the main text since it highlights the genes that are distinct between seawater and *A. kenti* microbiomes? I feel like it would be useful to have in the main text to compare the two sets of MAGs.

Figure S2 was not included as a main Figure as we felt that the vast numbers of significantly enriched genes were better interrogated by the reader within Table S2. However, considering the reviewer’s comment, this has now become Figure 4 within the manuscript.

- L317-318 – I love the section on different aspects of nitrogen cycling in corals. I was hoping to see more discussion of the Babbin 2021 paper (currently citation #10). The paper conducted nitrogen cycling rate measurements in Caribbean corals and coral-derived cultures of nitrogen cycling microorganisms. In particular they measured nitrate reduction (among other pathways). While this was done in corals in a different geographic region, it warrants further discussion in this section, and directly addresses the statement “functional validation of these largely anaerobic processes is required”.

We have expanded this discussion, so that we now state on lines 316-326, “*Although the distribution of several of these lineages across the *A. kenti* colonies was patchy (Figure 2; see Supplementary Note 4) and functional validation of these largely anaerobic processes within *A. kenti* is required, the genomic potential for anaerobic nitrogen metabolisms is consistent with previous research demonstrating the activity of nitrate reduction within Caribbean scleractinian corals and bacterial isolates from the coral holobiont ¹⁰. The greater number of lineages with complete assimilatory as opposed to reduction pathways (Figure 5), suggests low nitrogen levels and limited nitrogen loss from the *A. kenti* holobiont, in-line with previous observations of low denitrification rates in corals relative to assimilatory pathways or nitrogen fixation ⁶⁵.*

Thus the activity of these MAGs may contribute to A. kenti holobiont homeostasis, by retaining nitrogen within the system^{10,67}.

- L442 “suggesting a role for the microbiome in environmental adaptation” – The section on microbiome changes corresponding to water quality is interesting, and it may also have to do with the geographic location of the corals. It might be worth mentioning that geographic differences, which are often related to water quality, have been shown to impact the structure of the coral microbiome in previous studies using 16S rRNA gene sequencing (Williams et al 2022, <https://peerj.com/articles/13574/>; Becker et al 2023, <https://academic.oup.com/pnasnexus/article/2/9/pgad287/7259987>)

We have added these references to lines 427-428, where we now state, “Differences in coral and seawater microbiome community structure are often related to geographic distances and reef water quality⁸³⁻⁸⁵...”.

- L519-520 – The citations might be off. For example, citation 86 does not seem appropriate for the Cladocopium C15 genome. Additionally, for ease of reproducibility, I suggest you include the repository accession or link to the publicly available genomes. Sometimes it can be difficult to identify the location from a citation alone.

We apologise for this error; we now correctly refer to citation 18. We have added links to the publicly available genomes within the Supplementary Methods, on lines 50-52, and checked, then corrected, any misnumbered references throughout the rest of the text.

- L526 – What were your CoverM parameters? Default?

Yes, we have added, “using CoverM ‘make’ with default settings (v0.2.0-alpha7; <https://github.com/wwood/CoverM>)” to this sentence on line 538.

- L528 – For using UniteM, are the parameters the “—verysensitive”, etc.?

Yes, those are the parameters used for the binning tools with UniteM as outlined in the text. We have also added the functions used, so that lines 540-545 now read, “...input to the custom ensemble binning tool UniteM (v1.0.0; <https://github.com/dparks1134/UniteM/>), using the commands ‘bin’, ‘profile’, and ‘consensus’ to produce a non-redundant set of MAGs from the binning methods: GroopM2 (<https://github.com/timbalam/GroopM>)⁹³, MaxBin using both the 40 and 107 marker sets⁹⁴, and MetaBAT2 and MetaBAT1 using the settings --verysensitive --sensitive --specific --veryspecific --superspecific⁹⁵.”.

- L533 – For RefineM, were there specific cutoffs you used for the outliers, or was it default?

This was based on the default settings for GC content and tetranucleotide distance, we have updated the description so that on lines 545-548, we now state, “Contigs identified

as outliers based on GC content and tetranucleotide distance were removed from the MAGs using RefineM and the commands ‘scaffold_stats’, ‘outliers’, and ‘filter_bins’ (v0.0.24; <https://github.com/dparks1134/RefineM>; ⁹³) with default settings.”.

- L555 – Did you use a specific command within EnrichM? Which one? Any specific parameters?

EnrichM ‘annotate’ and ‘classify’ were used. On lines 570 and 574-575 we now state, “EnrichM ‘annotate’ (v0.4.15; <https://github.com/geronimp/enrichM>) was used to...”, and, “EnrichM ‘classify’ (v0.4.15) was used to determine the completeness of KEGG Modules (%) based on KO annotations...”.

- L555-556 – Did you generate the annotated predicted proteins of MAGs, *A. kenti*, and *C. goreau*? If so, how?

The predicted proteins of *A. kenti* and *C. goreau* were downloaded from the public repositories, along with the genomes (links now provided in the Supplementary Methods) and annotated with Cazy, KO, and Pfam databases with the MAGs using EnrichM. The predicted proteins within the MAGs were determined during the EnrichM ‘annotate’ workflow which parses several dependencies, including Prodigal if fasta files of predicted proteins are not supplied. This has been clarified on lines 570-571, where we now state, “EnrichM ‘annotate’ (v0.4.15; <https://github.com/geronimp/enrichM>) was used to annotate predicted proteins of all MAGs, and the publicly available predicted proteins of *A. kenti* and *C. goreau* (see Supplementary Methods)...”.

- L594 – Did you use a specific method for normalizing for different metagenome sizes? If so, what method?

This was achieved manually by scaling the mean coverages of MAGs to the smallest library size of the metagenomes (where, library size = number of reads multiplied by the average read length, in Gb). This detail has been added to lines 585-588, which now read, “The resulting table was normalised to account for differences in metagenome sequencing depth between samples, by scaling to the smallest library size (Gb; number of reads × average read length) and transformed to relative abundance.”.

- L598 and rest of paragraph – Can you provide the capscale models (and envfit) you used?

Indeed, the models have been exported from R as .rda file types and added as Supplementary Datasets, as per the following:

‘Island_origin_capscale_model.rda’, is the constrained ordination of *A. kenti* and seawater community composition based on the Island site origin of the samples, using the capscale function of vegan.

'Island_origin_capscale_model_envfit.rda', is the model testing the fit of the Island site origin metadata categories to the constrained capscale ordination of *A. kenti* and seawater community composition.

'A.kenti_WQ_capscale_model.rda', is the constrained ordination of *A. kenti* community composition based on the water quality categories, using the capscale function of vegan.

'A.kenti_WQ_capscale_model_envfit.rda', is the model testing the fit of the water quality categories to the constrained capscale ordination of *A. kenti* MAGs community composition.

- Methods general – will you provide scripts used in a repository or supplementary figure for publication?

All code that has not previously been published is available through GitHub at the respective links provided in the Methods and Supplementary Methods.

- L617 – which statistical tests?

This was in reference to the Mann Whitney U Test described in the previous sentences. On lines 636-639 we now state, “*The normalised relative abundances of the overall dereplicated MAGs were used to weight cluster frequencies by the relative abundance of the organisms encoding them, using water quality categories as factors for the Mann Whitney U Test (P-value < 0.05; Table S6).*”.

- Data availability – Does this BioProject include both raw sequence reads as well as the MAGs? Will the analytical code be available too?

Yes, both metagenomic reads and MAGs are included within BioProject PRJNA545004. As mentioned above, any code not previously published is available through GitHub at the links provided.

- L924 – “Metabolic overview of key modules” – How were these identified? What tool? Was this in the methods?

We apologise for this omission. This was determined using EnrichM, on lines 574-575 we now state, “*EnrichM ‘classify’ (v0.4.15) was used to determine the completeness of KEGG Modules (%) based on KO annotations.*”.

- L948 – Supplementary Table Captions – When I downloaded the supplementary Excel files, they had no clear titles. In the top of the Excel sheets or in the tab, could you clearly label, for example “Table S2”. If this could be done for all the supplementary details that would be helpful since I believe they get removed from the file names.

The Supplementary Table Captions have now been added to the top of the Excel sheets – we apologise this information was lost from the file names.

- Figure 2 – The Good to Poor water quality gradient seems misleading. Is it truly a gradient, or was it Good water quality at “S. marine”, medium water quality at “S. plume” and poor at “Coastal”. Similarly, were “N. Plume” and “Coastal” equally poor? Maybe three distinct colors there would be useful.

We have removed the colour gradient as requested and added the three categories ('Good', 'Moderate', and 'Poor' in-line with the GBRMPA terminology for the inshore reef Marine Monitoring Program) as three distinct colours.

- Figure 3 – Could you label E, F, and G with the different annotation databases?

We have added the annotation databases in the top right-hand corner of the PCoA plots in E, F, and G.

- Figure 3 – The gradient color scheme for taxa is somewhat hard to distinguish. Is there a reason it was chosen? Perhaps something more distinct would be useful unless it isn't critical to the story.

This colour scheme was chosen as an accessible colour-blind-friendly- palette. As the Reviewer 1 did not raise this as a concern, we have opted to leave it as is, however if the Editor feels this should be changed, we will be happy to do so.

- Figure 4 – Will you provide definitions for the energy metabolism acronyms?

These acronyms have now been defined in the Figure caption, added on lines 960-965 of the revised manuscript file.

- Figure 5 – For consistency, could you also label the different plumes by their water quality gradient designation?

We have added 'Good', 'Moderate' and 'Poor' to the colour key in what is now Figure 6.

- Supplementary Information:

- L49 – Any specific parameters for Seqpurge?

We have added, “...adaptors were removed using Seqpurge (ngs-bits/2018_11) with the flags -ncut 0 and -qcut 0,...”.

- L50 – It was unclear from the Robbins paper citation for Cladocopium C15 where to get the genome, though I did ultimately track it to reefgenomics.org? Maybe also cite that repository directly? If there is an accession number for it, add that too.

We have added the link to the *Cladocopium C15* genome on line 51.

- L50 – Same for *Cladocopium goreau*. The citation had a clear location, but the repository also has a citation that you can include: Chen, Yibi, González-Pech, Raúl A., Stephens, Timothy G., Bhattacharya, Debashish, and Chan, Cheong Xin(2019). Revised genome sequences and annotations of six Symbiodiniaceae taxa. The University of Queensland. Data Collection. <https://doi.org/10.14264/uql.2019.745>

We have added the link to the Data Collection DOI on line 52.

- L54-64 – I was confused why the reads were assembled with megahit, when in the body of the text it said that Spades was used (L524). Can you clarify what went on here?

Certainly, megahit was used for the rigorous quality control procedure due to its speed and memory-efficient processing of the largest metagenomes sequenced herein (80-100GB). Even with access to high performance computing with substantial memory allocation (1.5TB RAM), we could not assemble the deep-sequenced metagenomes with metaSPAdes prior to completing the quality control. However, as metaSPAdes has previously been shown to produce superior assemblies from Illumina data¹⁻³, we opted to use metaSPAdes for the metagenome-assembled genome reconstruction workflow. We compared the statistics of the metagenomic assemblies of the remaining microbiome reads obtained using both megahit and metaSPAdes, and indeed found that metaSPAdes resulted in longer contigs and higher N50 values compared to megahit.

On line 58 of the Supplementary Methods we now state, “*To improve processing times, the host and Symbiodiniaceae removed reads were assembled using megahit (v1.1.4)...*”.

1. van der Walt, A. J. *et al.* Assembling metagenomes, one community at a time. *Cold Spring Harbor Laboratory* 120154 (2017) doi:10.1101/120154.
2. Lapidus, A. L. & Korobeynikov, A. I. Metagenomic Data Assembly – The Way of Decoding Unknown Microorganisms. *Front. Microbiol.* 12, (2021).
3. Vollmers, J., Wiegand, S. & Kaster, A.-K. Comparing and Evaluating Metagenome Assembly Tools from a Microbiologist’s Perspective - Not Only Size Matters! *PLoS One* 12, e0169662 (2017).

We would like to thank the Reviewer’s once again for their thoughtful assessment of our manuscript.

REVIEWERS' COMMENTS

Reviewer #1 (Remarks to the Author):

Thank you for your thorough response! The manuscript is great. All my comments have been addressed.

Reviewer #2 (Remarks to the Author):

The authors did a great job addressing all of my revisions as well as the other reviewer's comments.

I noticed they chose not to adjust the color palette. I agree with this decision, especially since it is a color-blind friendly palette.

The only minor comment has to do with your discussion of the control in the supplement. After the other reviewer's comments regarding the DNA extraction control, I went and read that section more closely. I thought the mapping of control DNA to the MAGs was sound, though I do find it very peculiar that *Cutibacterium acnes* MAGs were not in the control. I think it would be worth adding that *C. acnes* is a commensal human skin bacterium.

I like that you additionally point out that *C. acnes* is a common contaminant in low biomass 16S rRNA studies. I was curious about this and looked at the citations 13, and 14 that you provided, and only 14 seemed to support that claim and did a nice job summarizing *Propionibacterium* findings in coral studies. Did you perhaps mean to include a different citation for 13?

Outside of that minor fix, I fully support accepting this paper for publication.

Response to Reviewers

Reviewer #1 (Remarks to the Author):

Thank you for your thorough response! The manuscript is great. All my comments have been addressed.

We would like to thank the Reviewer once again for their comments which improved the quality of the final manuscript.

Reviewer #2 (Remarks to the Author):

The authors did a great job addressing all of my revisions as well as the other reviewer's comments.

We would like to thank the Reviewer for the time spent on their thorough and constructive comments.

I noticed they chose not to adjust the color palette. I agree with this decision, especially since it is a color-blind friendly palette.

Thank you.

The only minor comment has to do with your discussion of the control in the supplement. After the other reviewer's comments regarding the DNA extraction control, I went and read that section more closely. I thought the mapping of control DNA to the MAGs was sound, though I do find it very peculiar that *Cutibacterium acnes* MAGs were not in the control. I think it would be worth adding that *C. acnes* is a commensal human skin bacterium.

We have added this detail in the Supplementary Information. The sentence now reads, "However, three of the A. kenti-specific MAGs with normalised relative abundances between 0.03 - 5.5%, were classified as Cutibacterium acnes (Actinobacteriota; Fitzroy_MAG20, Russell_MAG30, and Magnetic_MAG20; formerly of the genus Propionibacterium) by GTDB, a common skin bacterium that has been observed in 16S rRNA sequencing studies from at least 15 coral species including Acropora tenuis¹¹, and recognised as a contaminant in low microbial biomass coral microbiome studies¹²."

I like that you additionally point out that *C. acnes* is a common contaminant in low biomass 16S rRNA studies. I was curious about this and looked at the citations 13, and 14 that you provided, and only 14 seemed to support that claim and did a nice job summarizing *Propionibacterium* findings in coral studies. Did you perhaps mean to include a different citation for 13?

We appreciate the care the Reviewer has taken to improve our manuscript. This has been checked and the correct citation was given – the key information in this citation is presented in their Table S1, which lists Propionibacterium as one of the most abundant microbial taxa in the Coral Microbiome Database, represented across 15 coral hosts including A. tenuis, which may well represent A. kenti based on the reclassification of this species. We still believe any supposition that this bacterium is an important component of the A. kenti microbiome requires localisation of the microorganism within the coral host but we have briefly elaborated on the findings of this citation in the Supplementary Information (as shown in the above text; now citation 11).

Outside of that minor fix, I fully support accepting this paper for publication.

Thank you once again for your contribution to improving our manuscript.